# Evaluating the dendroclimatological potential of blue intensity on multiple conifer species from Tasmania and New Zealand

Rob Wilson[1,4], Kathy Allen[2], Patrick Baker[2], Gretel Boswijk[3], Brendan Buckley[4], Edward Cook[4], Rosanne D'Arrigo[4], Dan Druckenbrod[5], Anthony Fowler[3], Margaux Grandjean[1], Paul Krusic[6], Jonathan Palmer[7]

[1] School of Earth & Environmental Sciences, University of St. Andrews, UK
[2] School of Ecosystem and Forest Sciences, University of Melbourne, 500 Yarra Boulevard, Richmond 3121, Australia
[3] Tree-Ring Laboratory, School of Environment, The University of Auckland, Private Bag 92019, Auckland, New Zealand
[4] Lamont-Doherty Earth Observatory, Palisades, New York 10964, USA
[5] Department of Geological, Environmental, and Marine Sciences, Rider University, 2083 Lawrenceville Rd, Lawrenceville, NJ, 08648, USA
[6] Department of Geography, University of Cambridge, Cambridge, UK
[7] ARC Centre of Excellence in Australian Biodiversity and Heritage, School of Biological, Earth and Environmental Sciences, University of New South Wales, Sydney, NSW 2052, Australia

*Correspondence to*: Rob Wilson (rjsw@st-andrews.ac.uk)

**Abstract.** We evaluate a range of blue intensity (BI) tree-ring parameters in eight conifer species (12 sites) from Tasmania and New Zealand for their dendroclimatic potential, and as surrogate wood anatomical proxies. Using a dataset of ca. 10-15 trees per site, we measured earlywood maximum blue intensity (EWB), latewood minimum blue intensity (LWB) and the associated delta blue intensity (DB) parameter for dendrochronological analysis. No resin extraction was performed, impacting low-frequency trends. Therefore, we focused only on the high-frequency signal by detrending all tree-ring and climate data using a 20-year cubic smoothing spline. All BI parameters express low relative variance and weak signal strength compared to ring-width. Correlation analysis and principal component regression experiments identified a weak and variable climate response for most ring-width chronologies. However, for most sites, the EWB data, despite weak signal strength, expressed strong coherence with summer temperatures. Significant correlations for LWB were also noted, but the sign of the relationship for most species is opposite to that reported for all conifer species in the Northern Hemisphere. DB results were mixed but performed better for the Tasmanian sites when combined through principal component regression methods than for New Zealand. Using the full multi-species/parameter network, excellent summer temperature calibration was identified for both Tasmania and New Zealand ranging from 52% to 78% explained variance for split periods (1901-1950 / 1951-1995), with equally robust independent validation (Coefficient of Efficiency = 0.41 to 0.77). Comparison of the Tasmanian BI reconstruction with a quantitative wood anatomical (QWA) reconstruction shows that these parameters record essentially the same strong high-frequency summer temperature signal. Despite these excellent results, a substantial challenge exists with the capture of potential secular scale climate trends. Although DB, band-pass and other signal processing methods may help with this issue, substantially more experimentation is needed in conjunction with comparative analysis with ring density and QWA measurements.

## 1 Introduction

The range of variables that are now routinely measured from the rings of trees, including width, stable isotopes, multiple wood anatomical properties and density, has increased substantially in recent years (McCarroll et al. 2002; McCarroll and Loader, 2004; Drew et al. 2012; von Arx et al. 2016; Björklund et al., 2020). However, our knowledge of the climatic, environmental, and physiological processes that modulate the year-to-year variability of these different tree-ring parameters is still far from comprehensive.

Since the early seminal work of Fritts et al. (1965), a well-known rule of thumb for ring-width (RW) based dendroclimatology is that trees sampled near their high elevation or latitude treelines will be predominantly temperature limited, while at lower elevations or latitudes, moisture limitation becomes the primary driver of growth (Fritts 1976; Kienast et al. 1987; Buckley et al. 1997; Wilson and Hopfmüller 2001; Briffa et al., 2002; Babst et al. 2013; St. George 2014). Such targeted sampling is strategically vital in "traditional" dendroclimatology and robust reconstructions can be derived so long as tree-line sites are sampled where a single dominant climate parameter controls growth (Bradley 1999). However, the climatic influence on RW can be complex and there are many published studies where the relationship between RW and climate is shown to be temporally unstable and/or non-linear (Wilmking et al. 2020).

Ring density parameters, especially maximum latewood density (MXD), have been shown to provide substantially more robust estimates of past summer temperature compared to RW (Briffa et al., 2002; Wilson and Luckman, 2003; Esper et al., 2012; Büntgen et al., 2017; Ljungqvist et al., 2019). Density data may also retain a strong temperature signal at elevations below the upper treeline, minimising the non-linear influence of a changing tree-line elevation through time (Kienast et al. 1987). The use of ring-density variables from lower elevation or latitude sites to reconstruct past hydroclimate is rare (Camarero et al. 2014, 2017; Cleaveland 1986; Seftigen et al. 2020) and is clearly an area demanding further attention.

The reconstructive value of tree ring stable isotopes (carbon and oxygen) appears to be less constrained for sites where climate does not limit growth and substantial potential exists from mid-latitude regions where traditional dendroclimatological approaches are less reliable (McCarroll and Loader, 2004; Loader et al. 2008; Young et al. 2015; Loader et al. 2020; Büntgen et al. 2021). However, within the mechanistic framework of stable isotopes, there is still much to explore regarding the complex associations between fractionation and climate for different species and across different ecotones.

The use of quantitative wood anatomical (QWA) parameters for dendroclimatology has gained traction in recent years due to improvements in measurement methodologies allowing for the development of well-replicated chronologies for multiple different anatomical variables (Drew et al. 2012; von Arx et al. 2016; Prendin et al., 2017; Björklund et al. 2020). The

strength of relationships between climate parameters and wood anatomical properties such as latewood cell wall thickness,
tracheid radial diameter and microfibril angle is comparable to and can be stronger than maximum latewood density (Yasue
et al., 2000; Wang et al., 2002; Panyushkina et al., 2003; Fonti et al., 2013; Allen et al. 2018).

Despite the strong climate signal often noted in such non-RW tree-ring parameters, their procurement is expensive, often
requires specialised equipment and experience, and is time consuming. Consequently, there are substantially less published
data available for inspection and assessment. In recent years, blue intensity (BI) has been championed by many groups as a
cheaper surrogate for maximum latewood density (Björklund et al., 2014a/b; Rydval et al., 2014; Wilson et al., 2014; Kaczka
and Wilson 2021). In its common usage, BI measures the intensity of the reflectance of blue light from the latewood of
scanned conifer samples so that a dense (dark) latewood would result in low-intensity values. MXD and BI essentially
measure similar wood properties. Most studies that have directly compared MXD and latewood BI show no significant
difference in the climate response of the two parameters (Wilson et al., 2014; Björklund et al 2019; Ljungqvist et al., 2019;
Reid and Wilson 2020). Though the acceptance of BI in dendrochronology was initially slow after the publication of the
original concept paper (McCarroll et al. 2002), over the past decade many BI-based studies have been published (Kaczka and
Wilson 2021). These studies have examined the use of BI as an ecological and climatological indicator in a variety of conifer
species from several locations around the Northern Hemisphere (Campbell et al., 2007, 2011; Helama et al., 2013; Rydval et
al., 2014, 2017, 2018; Björklund et al., 2014a/b; Wilson et al., 2014, 2017a, 2017b, 2019; Babst et al., 2016; Dolgova, 2016;
Arbellay et al., 2018; Buras et al., 2018; Fuentes et al., 2017; Kaczka et al., 2018; Wiles et al., 2019; Harley et al. 2021;
Heeter et al. 2020; Reid and Wilson 2020; Davi et al. 2021).

Only three studies that utilise BI data south of 30°N have been published. Buckley et al. (2018) explored the potential of
reflectance parameters from the tropical conifer Fujian cypress (Fokienia hodginsii) from central Vietnam and found a
significant positive relationship between earlywood maximum BI and December-April maximum temperature. Although a
spring/early summer temperature signal is extant in Northern Hemisphere conifer minimum density data from temperature
limited sites (Björklund et al. 2017), correlations are generally not as strong as the earlywood results detailed by Buckley et
al. (2018). In the Southern Hemisphere, Brookhouse and Graham (2016) measured latewood BI from Errinundra plum-pine
(Podocarpus lawrencei) samples taken from the Australian Alps and identified a strong inverse (r = -0.79) relationship with
August-April maximum temperatures, suggesting substantial potential for this species if long-lived specimens could be
found. Finally, Blake et al. (2020) recently explored the climate signal in BI parameters measured from Silver pine (Manoao
colensoi) samples growing on New Zealand's South Island and found strong significant relationships between both
earlywood and latewood BI parameters and summer temperatures. Although the sign (positive) of the earlywood BI
relationship with temperature agreed with results detailed in other studies (Björklund et al. 2017; Buckley et al. 2018), the
latewood relationship was inverse to that detailed for Northern Hemisphere conifers (Briffa et al. 2002) and observed by
Brookhouse and Graham (2016). This difference in latewood response begs the intriguing question as to whether some

Southern Hemisphere conifers may have evolved differently from their Northern Hemisphere counterparts, resulting in a different anatomical and physiological response to climate.

Here we expand upon the pilot studies of Brookhouse and Graham (2016) and Blake et al. (2020) and explore the climate signal of BI parameters from several key conifer species from Tasmania and New Zealand. To minimise nomenclature confusion, we refer to the different BI parameters as earlywood blue intensity (EWB) and latewood blue intensity (LWB). Based on ecophysiological theory (Buckley et al. 2018) we posit that EWB, derived from maximum intensity values of the whole-ring reflectance spectrum, essentially provides a surrogate for mean lumen size of the earlywood cells, while LWB, derived from minimum reflectance values, reflects the relative density (i.e. the proportion of cell wall to lumen area) of the darker latewood cell walls. We further suggest these reflectance measures are useful surrogate measures of mean tracheid diameter and cell wall thickness, which are proven to be excellent proxies of past climate (Allen et al. 2018; Björklund et al. 2019) but are laborious and expensive to measure directly. As well as undertaking a dendroclimatic assessment of multiple BI parameters from different Australasian conifers, our analysis will also identify which species would be a good focus for further BI and QWA measurement in the future. Improving terrestrial-based estimates of past temperature in the land-limited Southern Hemisphere (Neukom et al. 2014) will only be achieved by enhancing the strength of the calibrated signal that until recently has been characterized solely by ring-width data which generally express a weak temperature signal.

| Site Name | Site code | Common name | Species | Latitude (S) | Longitude | Elevation (m) | No of series | No of trees | full period | period ≥3 series |
|---|---|---|---|---|---|---|---|---|---|---|
| **TASMANIA** | | | | | | | | | | |
| Race Spur | RCS | Celery Top pine | *Phyllocladus aspleniifolius* | 41.29 | 145.44 | 500-550 | 16 | 14 | 1788-1995 | 1795-1995 |
| L. Mackenzie | MCK | Pencil pine | *Athrotaxis cupressoides* | 41.41 | 146.23 | 1116 | 15 | 15 | 1771-2007 | 1780-2007 |
| Cradle Mountain | CM | Pencil pine | *Athrotaxis cupressoides* | 41.40 | 145.57 | 1050 | 15 | 15 | 1787-2001 | 1789-2001 |
| Mt Weld West / Trout Lake | MWWTRL | King Billy pine | *Athrotaxis selaginoides* | 43.00 | 146.34 | 950 | 17 | 9 | 1781-1998 | 1785-1998 |
| Mt Read - KBP | MRD | King Billy pine | *Athrotaxis selaginoides* | 41.50 | 145.32 | 900 | 13 | 6 | 1770-2010 | 1778-2010 |
| Mt Read - HP | MHP | Huon pine | *Lagarostrobos franklinii* | 41.50 | 145.32 | 1000 | 22 | 16 | 781-2002 | 1238-2001 |
| John Butters Power Station (King River) | BUT | Huon pine | *Lagarostrobos franklinii* | 42.15 | 145.30 | 60 | 10 | 10 | 1773-2008 | 1798-2008 |
| | | | | | | | | | | |
| **NEW ZEALAND** | | | | | | | | | | |
| Puketi | PKL | NZ Kauri | *Agathis australis* | 35.15 | 173.45 | 180 | 13 | 10 | 1674-2001 | 1737-2001 |
| Huapai | HUP | NZ Kauri | *Agathis australis* | 36.48 | 174.3 | 100 | 17 | 13 | 1664-2007 | 1723-2006 |
| Flagstaff | FLC | NZ Cedar | *Libocedrus bidwillii* | 42.30 | 171.43 | 280 | 12 | 7 | 1774-2004 | 1776-2004 |
| Ahaura | AHA | Silver pine | *Manoao colensoi* | 42.23 | 171.48 | 244 | 12 | 12 | 1750-2012 | 1750-2012 |
| Doughboy, Stewart Island | DPP | Pink pine | *Halocarpus biformis* | 46.59 | 167.43 | 230 | 20 | 12 | 1767-2010 | 1777-2010 |

**Table 1: Chronology information for the seven Tasmanian and five New Zealand sites used in the study (see Figure 1).**

**2 Data and Methods**

Four tree species from Tasmania and four from New Zealand were targeted for analysis (Figure 1, Table 1) representing conifer species that have not only been the focus of previous dendrochronological studies, but each has the potential to produce climate proxy records substantially greater than 1000 years in length. Until recently, RW data were used for most

Australasian dendroclimatological studies, with calibration results never exceeding 40-45% explained variance. In Tasmania,
the strongest calibration results for summer temperatures had been obtained using high elevation Huon pine (*Lagarostrobos*
*franklinii* - Buckley et al. 1997; Cook et al. 2006) although some coherence was also found for Pencil pine (*Athrotaxis*
*cupressoides*) and King Billy pine (*Athrotaxis selaginoides* - Allen et al. 2011; Allen et al. 2017). The study sites (Table 1)
for Pencil pine (MCK and CM) and King Billy pine (MWWTRL and MRD) are located close to the upper timberline limit of
these species and growth is expected to be controlled mostly by summer temperatures. Likewise, the high elevation Huon
pine (MHP) site is also close to the upper treeline where summer temperature is the dominant response (Buckley et al. 1997).
However, BUT is located at the lower end of the Huon pine elevational range within a riparian environment so temperature
limitation is unlikely in a traditional sense. However, Drew et al (2012) identified strong summer temperature signals in
latewood QWA data for this site. Celery Top (*Phyllocladus aspleniifolius*) RW data, however, express a complex non-linear
relationship with climate along its species' elevational range and have not been used for dendroclimatic reconstruction
(Allen et al. 2001). By contrast, summer temperature calibration experiments performed on measurement series of several
wood anatomical properties (e.g. tracheid radial diameter, cell wall thickness and microfibril angle), as well as RW and ring
density, from these same species, have shown substantial improvement over RW alone (Allen et al. 2018), although these
QWA data have been more useful for hydroclimate reconstructions (Allen et al. 2015a/b). In New Zealand, RW-based
summer temperature reconstructions have been developed from NZ Cedar (*Libocedrus bidwillii* - Palmer and Xiong 2004),
Silver pine (*Manoao colensoi* - Cook et al. 2002, 2006) and Pink pine (*Halocarpus biformis* - D'Arrigo et al. 1996, Duncan
et al. 2010) although ring density (Xiong et al. 1998 – Pink pine) and BI (Blake et al. 2020 – Silver pine) measured from the
earlywood have produced stronger results. For this study, we specifically measured BI from samples used in previous,
mostly RW-based dendroclimatic, studies where summer temperature was found to be the dominant climate signal - at least
for NZ Cedar, Silver pine and Pink pine. The sites for these three New Zealand species are close to their southern
(latitudinal) limits (especially the Stewart Island Pink pine site) which is thought to compensate, to some degree, for their
modest elevational range (Table 1). Kauri (*Agathis australis*) is the longest-lived tree species in Australasia (Boswijk et al.
2014) but only a few sites of reasonably mature trees exist. Previous analyses have identified a complex mixed response to
both temperature and precipitation through the growing season (Buckley et al 2000, Fowler et al, 2000). However, it is
notable that Kauri RW data express a strong stable relationship with indices of the El Nino Southern Oscillation (Cook et al.
2006; Fowler et al. 2012).

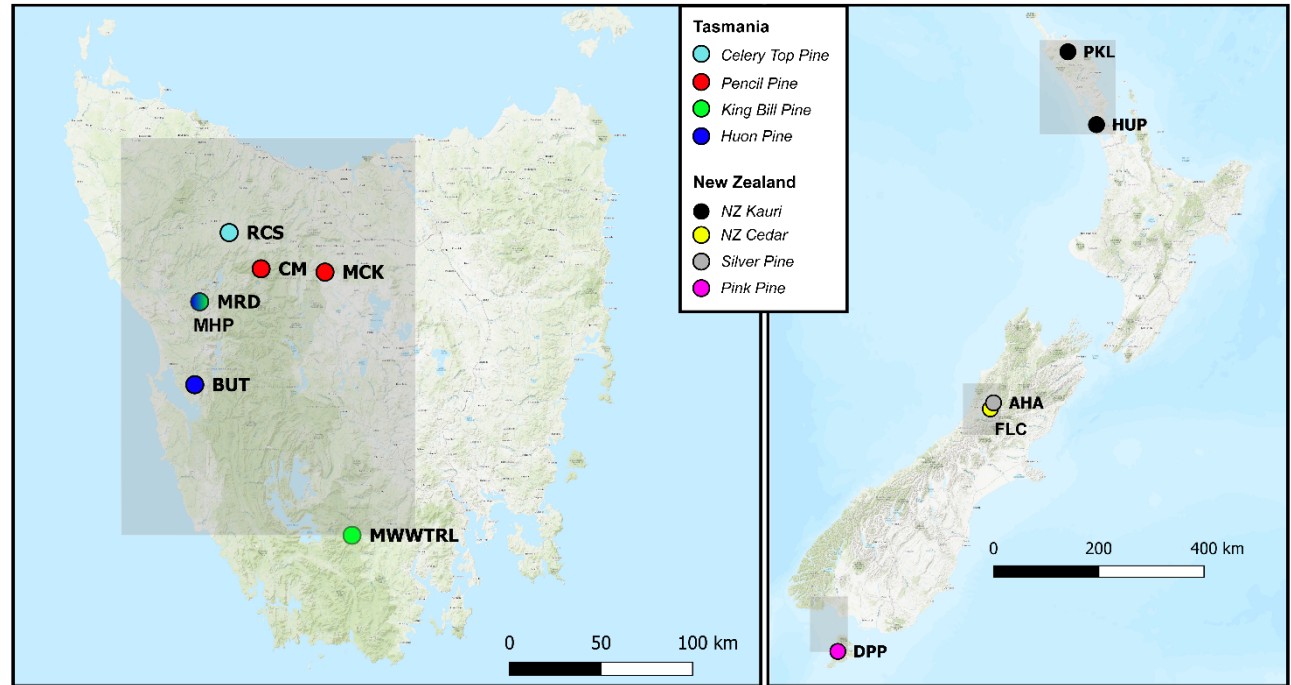



Figure 1: Location map (basemap ESRI 2021) of the tree-ring sites used in this study (see Table 1). Also indicated (grey boxes) are the regional domains of the gridded CRU TS 4.03 temperature and precipitation data (Harris et al., 2014) used for analyses. Tasmania: 145-147ºE / 41-43ºS; New Zealand: North: 173-175ºE / 35-37ºS; Central: 171-172ºE / 42-43 ºS ; South: 167-168ºE / 46-47ºS.

In this study, we utilised tree cores sampled over the past three decades that has been prepared for RW measurement. Considering the focus of this study is to assess the potential of BI parameters for enhancing dendroclimatic reconstruction, and the fact that the samples were already mounted, no resin extraction was performed except for the Silver pine AHA site (see Blake et al. 2020 for details). As many of the species are resinous by nature, this immediately imposes a potential problem for measuring BI data, because any inhomogeneous resin-related discolouration will impact intensity values (Rydval et al., 2014; Björklund et al., 2014a/b; Wilson et al. 2017b; Reid and Wilson 2020). Consequently, as the high-frequency signal will only be minimally affected by discolouration (Wilson et al. 2017a), all analyses for this proof-of-concept study will utilise only the high-pass fraction of the chronologies.

The mounted samples were re-sanded using fine grade (> 600 grit) sandpaper to remove decadal markings. Samples were scanned at multiple institutions using different scanners and a range of resolutions from 1200 to 3200 DPI. RW and BI data were generated using CooRecorder (Cybis 2016, http://www.cybis.se/forfun/dendro/index.htm) except for AHA

(WinDendro – see Blake et al. 2020). Regardless of image resolution, the CooRecorder BI generation "window" was set to
roughly equate to two-thirds width of the sample while the window depth encompassed either the latewood or earlywood for
each ring. The BI data were extracted following the method detailed in Buckley et al. (2018). For LWB, mean reflectance
values were taken from the lowest 15% of the darkest pixels, while for EWB the mean of the brightest 85% of the pixels was
used. Despite many of the samples being substantially older, most samples were measured only back into the 17th or 18th
centuries (with site MHP (Table 1) being an exception), providing enough data to ensure robust calibration and validation
over the instrumental period and to allow comparison with a temperature reconstruction from Tasmania based on QWA data
(Allen et al. 2018). Parameters generated for analysis were RW, EWB and LWB. As the study focuses only on the high-
frequency signal extant in the tree-ring data, the LWB data were not inverted as is the norm in Northern Hemisphere studies
using data generated in CooRecorder (Rydval et al. 2014).

Perhaps the greatest limitation for BI data parameters is that any colour changes that do not represent year-to-year changes in
wood anatomical features such as lumen size and cell wall thickness will impose a colour-related bias in the intensity
measurements. Examples of non-anatomically related colour changes are those associated with the heartwood/sapwood
transition, sections of highly resinous wood, or fungal staining. Björklund et al. (2014) proposed a statistical procedure that
could correct for such colour changes. This procedure subtracts the LWB reflectance value from the EWB data producing a
delta parameter (hereafter referred to as Delta BI - DB). Theoretically, DB should correct for common colour change biases
between heartwood and sapwood and even resinous zones within the wood. To date, DB has been utilised successfully in
only a few studies (Björklund et al., 2014a/b; Wilson et al., 2017b; Fuentes et al. 2017; Blake et al. 2020; Reid and Wilson
2020). As no resin extraction was performed (except site AHA, Table 1) and all the species used for this study express a
colour change from heartwood to sapwood, DB data will also be examined to explore its high-frequency dendroclimatic
potential.

For some of the studied species, the heartwood/sapwood transition colour change is very sharp and pronounced in
reflectance values (Figure A1), and inflexible detrending options could impose a systematic bias in the resultant detrended
indices. As an extreme example, the heartwood/sapwood transition of the EWB raw mean non-detrended chronology for the
CM Pencil pine site (Figure A2) cannot be tracked well with cubic smoothing splines (Cook and Peters 1981) of 200, 100 or
even 50 years respectively. This is not surprising given that the smoothing spline, operating as a symmetric digital filter, is
not well suited for dealing with abrupt changes in time series such as that observed in the CMewb chronology. In fact, the
bias of low (pre-transition) and high (post- transition) index values are only minimised when a flexible 20-year spline is used
because it better adapts to the observed discontinuity. However, this adaptability comes at the cost of losing potentially
valuable >20-year variability in the time series. This is clearly undesirable and better ways of modelling and removing such
discontinuities without the unwanted loss of lower-frequency variability are needed (see later discussion).  Although less
flexible splines could be used for other species with a gradual or minimal colour change from heartwood to sapwood (Figure
A1), a consistent approach to detrending was deemed prudent and therefore a 20-year spline was used for all datasets.
The mean interseries correlation statistic (RBAR) is utilised to assess how many series are needed to attain an Expressed
Population Signal value of 0.85 (Wigley et al. 1984; Wilson and Elling 2004). Previous research has shown that the common
signal expressed by BI data can be rather weak (Wilson et al. 2014, 2017a/b, 2019; Kaczka et al. 2018; Wiles et al. 2019).
We explore this phenomenon further with this multi-parameter/species network by using the coefficient of variation to help
understand relative internal variance and covariance of the parameter chronologies.
The climate signal expressed in the individual chronologies was initially explored using simple correlation analysis against
monthly gridded (see Figure 1 for locations) CRUTS 4.03 temperature and precipitation data (Harris et al. 2014) for the
periods 1902-1995, 1902-1950 and 1951-1995. Although the CRU TS data start in 1901, 1902 was the initial start year as
correlations were performed over 20 months including the previous growing season while 1995 reflects the final common
year for all tree-ring datasets (Table 1). The climate data were similarly detrended as the tree-ring data to ensure consistency.
Unsurprisingly, as most of the study sites are located in temperature limited upper tree-line locations, correlations with
monthly precipitation were weak, variable and temporally unstable for all species/parameter chronologies studies. The
results are presented in the Appendix but are not discussed further (see Table A4a-d).
Principal component analysis (PCA) was used on varying subsets of chronologies for each region (i.e. all chronologies of the
same parameter, or all parameters from a single species) to reduce the data to a few modes of common variance. Principal
components that had both an eigenvalue > 1.0 and correlated significantly (95% C.L.) with the target instrumental data were
entered into a stepwise multiple regression and calibrated against a range of seasonal temperatures. For New Zealand, the
three CRU TS 4.03 grid boxes (Figure 1) were averaged to create a countrywide mean series. This was justified as the three
inter-grid boxes mean correlation values between all tested seasons was 0.93 (STDEV = 0.01) suggesting there is a strong
common temperature signal between North Island and southern South Island. PCA was also utilised to ascertain the optimal
season for dendroclimatic calibration using the full chronology network for each country as well as exploring seasonal
differences between parameters and species. Analyses were performed over the common period of all tree-ring and climate
data (1901-1995) as well as early (1901-1950) and late (1951-1995) period calibration and verification. The Coefficient of
Efficiency (CE - Cook et al., 1994) was used to validate the regression-based climate estimates.

## 3 Results and Discussion

### 3.1 Chronology variability and signal strength

Wilson et al. (2014), using upper tree-line temperature-sensitive spruce samples from British Columbia, noted lower mean coefficient of variation (CV) values for LWB (0.05) compared to RW (0.28) and MXD (0.19). Common signal strength was strongest for the MXD data (RBAR = 0.42) while RW and LWB expressed similar but lower values (0.30). For the Australasian detrended data, overall, RW data express higher relative variance (mean CV = 0.13) followed by DB (0.07), LWB (0.03) and EWB (0.02 – Figure 2a). The range in values for RW (0.09 – 0.17) and DB (0.04 - 0.13) are greater than LWB (0.02 - 0.06) although there is overlap in the range of DB and LWB. The EWB data express a significantly narrower range (0.01 - 0.02). RBAR values for the four different parameter groups generally return a stronger common signal for RW (mean = 0.33) compared with EWB (0.14), LWB (0.16), and DB (0.15 – Figure 2b). Therefore, following traditional methodologies to assess signal strength, more BI series are needed than RW to attain a robust chronology. On average across all sites, to attain an EPS value of at least 0.85 (Wigley et al. 1984), 14 series would be needed for RW, while 44, 47 and 58 series would be needed for EWB, LWB and DB respectively. This weaker common signal of the BI parameters has been noted before (Wilson et al. 2014, 2017a/b, 2019; Kaczka et al. 2018; Wiles et al. 2019; Blake et al. 2020) and is also noted in QWA data from Tasmania (Allen et al. *in prep*). The common signal is particularly weak for Celery Top and Kauri (EWB) and Pink pine and Kauri (LWB and DB – see Table A1 for detailed values).

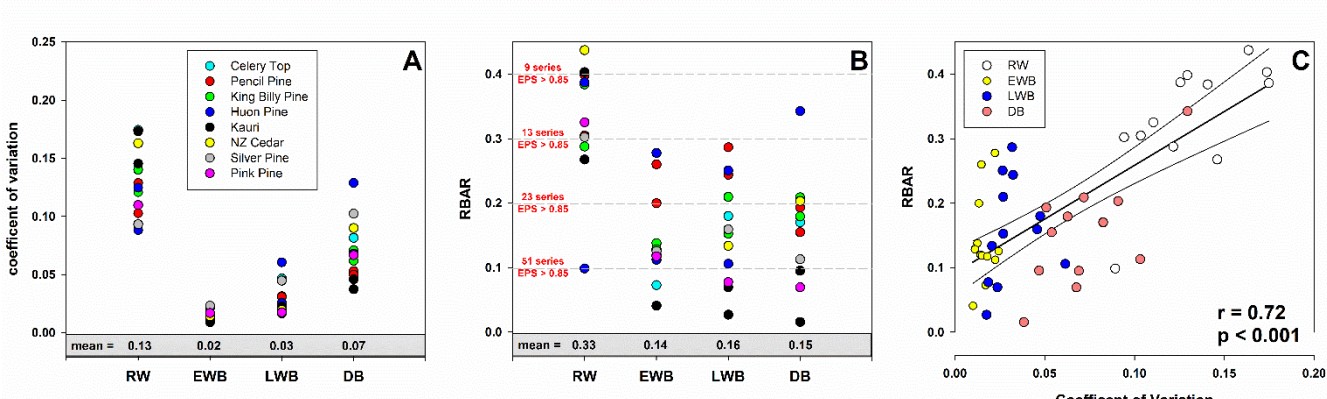

**Figure 2: A. Coefficient of variation (CV) of the 20-year spline detrended chronologies; B: mean inter-series correlation (RBAR) of the 20-year spline detrended series. Horizontal dashed lines denote the number of series needed for that particular RBAR value to attain an EPS of 0.85; C: Scatter plot of CV versus RBAR with linear regression.**

A scatter plot of the CV and RBAR data (Figure 2c) suggests that the common signal expressed by these chronologies is partly a function of the relative variance of the time-series (r = 0.72, p < 0.001). Although the range in RBAR values for the EWB and LWB data suggests some uncertainty in this observation (see also Table A1), these results imply that the relatively

low variation of values around the mean for the BI parameters suggests that any anomalous colour staining on the wood that
does not reflect the true wood properties being measured could have a substantial impact on the chronology common signal.
However, it should be emphasised that a weak common signal and low EPS value does not necessarily result in a weak
climate signal (Buras 2017).
**3.2 Climate response**
The strength of correlations between the RW chronologies and mean monthly temperatures vary in sign and strength across
species. Over the full 1902-1995 period (Table 2), the Tasmanian MWWTRL (King Billy pine) and MHP (high elevation
Huon pine) sites express significant positive correlations with September-February and January-February respectively,
which are broadly time stable (Table A3a). RCS (Celery Top pine), MCK (Pencil pine) and MRD (King Billy pine) show
inverse correlations with late summer temperatures of the previous year. Of the New Zealand sites, PKL (Kauri) has negative
correlations for many months from winter through to the summer, while AHA (Silver pine) and DPP (Pink pine) correlate
positively with December-April and September- November.

Correlations between the EWB chronologies and mean temperatures are surprisingly consistent for most sites although
correlations for RCS (Celery Top pine) and BUT (low elevation Huon pine) are weak. Almost all site chronologies correlate
positively with the summer months for the current season – December through to March (Tasmania) and December-February
(New Zealand). The King Billy pine and Kauri sites express narrower (MMWTRL, MRD, PKL, and HUP) response
windows while DPP (Pink pine) is wider (Table 2). Although these relationships appear generally time stable, the Tasmanian
sites correlate more strongly with the narrower January-February season for 1902-1950 compared to the later post-1951
period (Table A3b). Significant correlations with winter and prior year temperatures are weaker and less consistent than for
current spring/summer. Overall, the consistent and strong correlations of EWB with summer temperatures are extremely
encouraging and show great promise for enhancing RW-based temperature reconstruction for both regions.

Significant relationships between LWB and summer and early Autumn temperatures are generally noted, although the results
are less consistent than those for EWB. Both RCS (Celery Top) and high elevation Huon pine (MHP) express negative
correlations that are in line with the positive MXD/temperature relationships noted in the Northern Hemisphere as the LWB
data are not inverted. Excluding MMWTRL (King Billy pine) and HUP (Kauri), which do not have any significant
correlations with temperature in the growing season, all the LWB chronologies express positive correlations with summer
and early autumn temperatures. This antithetic behaviour is not a new observation and has been noted by Drew et al. (2012),
O'Donnell et al. (2016), Blake et al. (2020) for latewood anatomical parameters and LWB data, but these new results suggest
that this physiological phenomenon is not based on a chance occurrence of a single species and is consistent between several
Australasian conifer tree species (Pencil pine, Huon pine (low elevation), Kauri, NZ Cedar, Silver pine and Pink pine). Blake

et al. (2020) explained the inverse LWB relationship as a reduction in the duration of secondary cell wall thickening in warmer years. Such "emergent" surprising results (Cook and Pederson 2011) clearly need further research and testing.

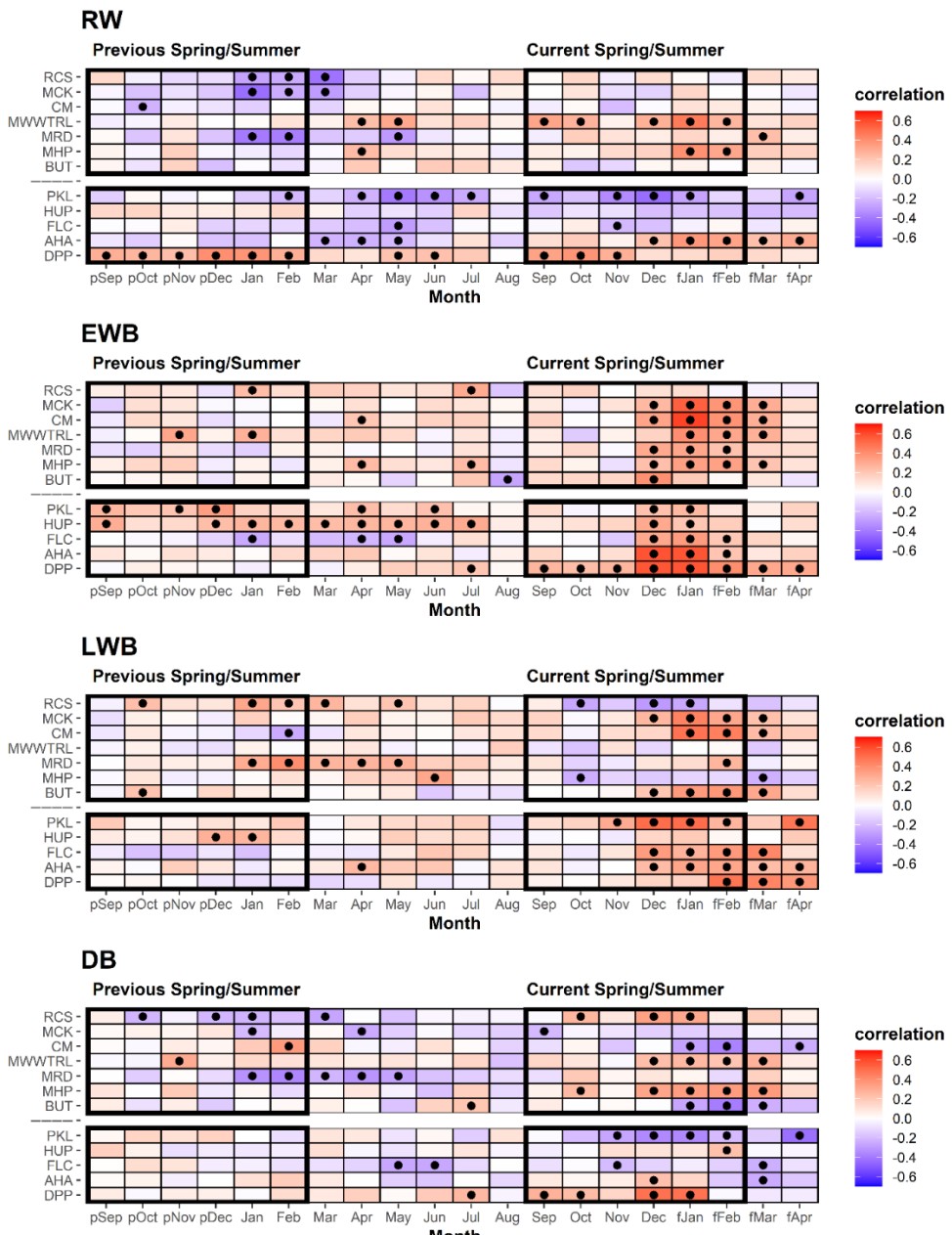

**Table 2: Correlation response function analysis results for the different TR parameter chronologies with CRU TS temperatures. Analysis undertaken over the 1902-1995 period (see supplementary figure S3 for correlations for split periods 1902-1950, 1951-1995). The upper block is for the Tasmanian sites, while the lower block is New Zealand. See Table 1 for site code names and species. Black dots denoted correlations significant at the 95% C.L.**


The DB chronologies express a range of responses to temperature that are all generally weaker than for EWB and LWB
(Table 2). Significant positive correlations with summer temperatures are found for RCS (Celery Top), MMWTRL (King
Billy pine), MHP (Huon pine), and DPP (Pink pine). HUP (Kauri) and AHA (Silver pine) also express some weak positive
summer temperature coherence. Negative correlations are noted for CM (Pencil pine), BUT (low elevation Huon pine), PKL
(Kauri) and FLC (NZ Cedar). However, many of these correlations are not temporally stable when compared over the 1902-
1950 and 1951-1995 periods (Table A3d). Current theory suggests that DB should perform well when EWB and LWB
parameters are weakly correlated and express different earlier and later seasonal climate responses (Björklund et al., 2014).
However, the results herein indicate that this simple hypothesis does not consistently apply in this multi-species study. For
example, the EWB and LWB data for the Pink pine DPP site express different early (Sep-April) and late (Feb-Apr) seasonal
responses with temperatures (Table 2), but still show a reasonably high inter-parameter correlation (0.60, Table A2) although
this is partly expected as the response windows overlap. However, the DB data still expresses a significant and strong
response with summer temperatures, although marginally weaker than the EWB response. On the other hand, DB for the
Pencil pine sites (MCK and CM) behaves more like conifers in the Northern Hemisphere (Björklund et al., 2014; Wilson et
al. 2017b), with significant correlations noted for both EWB and LWB with summer temperatures, but, likely due to the high
inter-parameter correlation (0.57 and 0.68), the DB data express weak, or even inverse correlations with summer
temperatures. Overall, the DB results are mixed and disappointing. This parameter theoretically could minimise the colour
bias of the darker to lighter colour heartwood/sapwood transition (Figure A1) but, for the data used herein, as the high-
frequency signal often portrays a mixed or weak signal with temperature, it suggests that the DB parameter might not be a
valid approach to address the heartwood/sapwood transition bias. These results suggest that alternative approaches to using
DB may need to be explored to minimise the impact of the heartwood/sapwood change noted in most of the species used in
this study.
**3.3 Parameter and species-specific principal component calibration tests**
The previous section detailed that temperature is the predominant climate signal expressed across the Tasmanian and New
Zealand RW and BI data studied herein (Figures 3, A3a-d). Only weak coherence with precipitation was found (Table A4a-
d). To further explore the climate response, principal component regression calibration (1901-1995) experiments with
seasonal temperature were performed to ascertain which combination of BI parameters and species express the strongest
climate signal and therefore should be the focus for future research – including refined BI measurement and/or QWA
measurement.

For Tasmania, the PCA identifies three (RW), two (EWB), two (LWB) and two (DB) principal components respectively.
Each BI parameter PC regression explains > 40% of the temperature variance while RW is substantially weaker at 21%
(Figure 3a). Both EWB and LWB explain 43% of the December-February and January-March variance respectively - these

seasons being biologically logical with respect to the earlier seasonal start for EWB and later end for LWB. Despite the site-specific DB data correlating with temperature more weakly than EWB and LWB (Table 2), their multivariate combination calibrates better (48%) with January-March temperatures. Although this is an encouraging result as DB may theoretically correct for colour related biases, the mix of positive and negative zero-order correlations with temperature (Table 2) suggest that some caution will be needed if such data are used to capture more secular scale information.

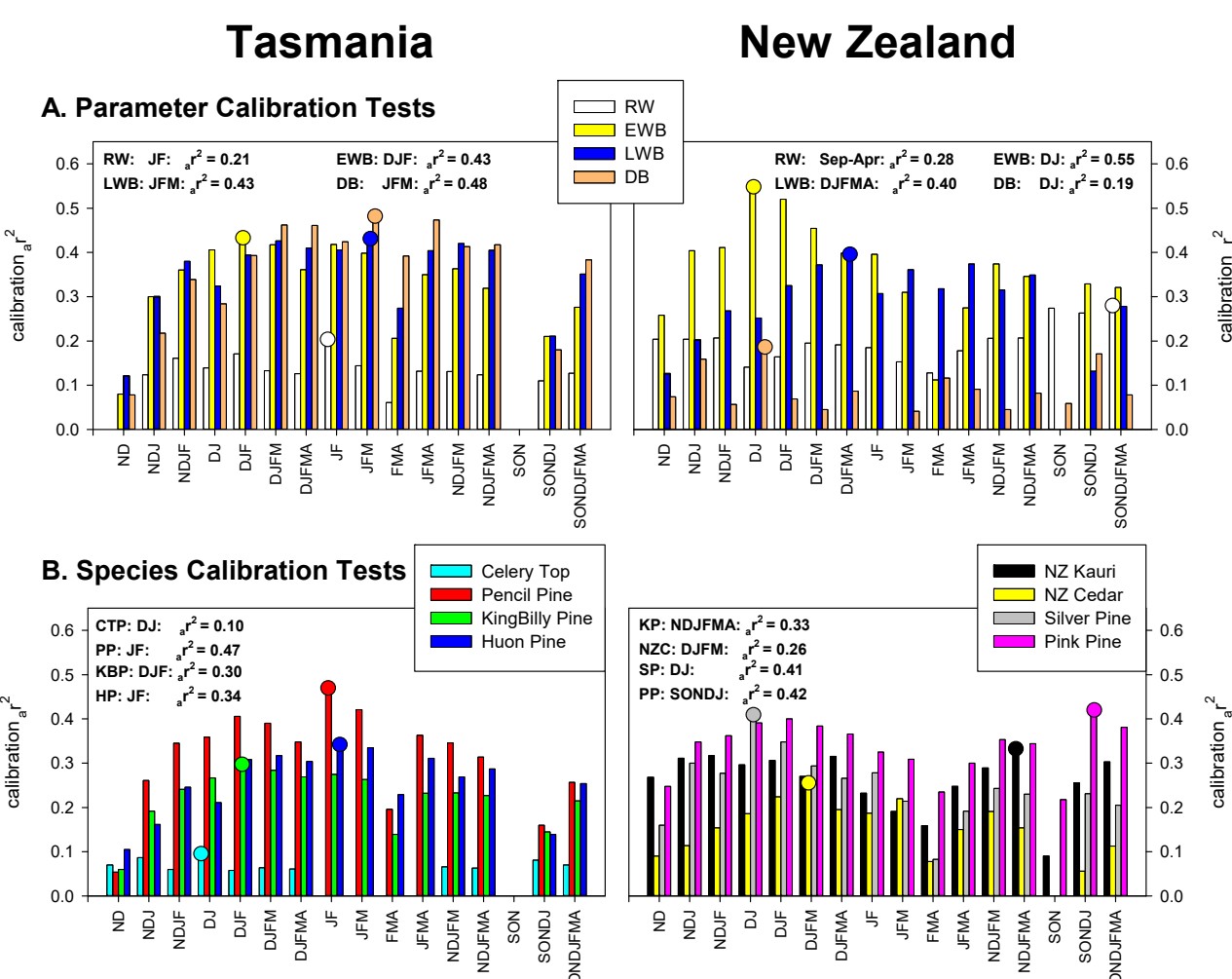

**Figure 3: PC regression calibration (1901-1995) experiments for parameters (all species) (A) and species (all variables) (B). A range of temperature seasonal targets are used with the strongest seasonal calibrations highlighted with circles.**

For New Zealand, PCA identifies 3, 2, 2 and 3 significant principal components for RW, EWB, LWB and DB respectively. EWB calibrates very strongly (55%) with December-January temperatures while LWB explains 40% of the broader

December-April season (Figure 3a). Alone, RW explains 28% of the temperature variance but for a broad September-April season which reflects the variable site-specific responses of PKL, AHA and DPP (Table 2). The DB data calibrate poorly explaining only 19% of the December-January temperature variance.

Of all the species tested, Tasmanian Pencil pine returns the strongest calibration (47%) with January-February temperatures (Figure 3b) although New Zealand Silver pine and Pink pine also calibrate reasonably with 41% (December-January) and 42% (September-January). It should be noted that two Pencil pine sites were used (Table 1) compared to only one each for Silver pine and Pink pine which likely will influence these results. King Billy pine, Huon pine and Kauri explain 30% (December-February), 34% (January-February) and 33% (November-April) respectively of the temperature variance with New Zealand cedar still showing some reasonable coherence (26%) for December-March. Celery Top is the weakest species explaining only 10% of the December-January temperature variance.

**3.4 Region-wide calibration and validation**

A multi-site, multi-species approach to dendroclimatology can improve overall calibration even if some of the sampled sites and species are not located close to climate limited treeline ecotones (Alexander et al. 2019). Herein we have an opportunity to pool all the data for each country to create combined multi-species and multi-parameter regional reconstructions. As the optimal season for calibration varies as a function of species and parameter (Figure 3), initial PC regression experiments, using all chronologies from each of the two regions, were performed. For each of these models, all PCs with an eigenvalue > 1.0 were entered into the regression model. January-February (JF) temperature was identified as the overall optimal season for Tasmania while December-January (DJ) provided the strongest calibration for New Zealand. Forcing all variables into the PC regression model also provides an opportunity to identify the importance of each species parameter towards the development of regional reconstructions. The beta weights (Cook et al. 1994) from the regression modelling (Table 3) clearly show the strong influence of the EWB parameters in the multiple regression model, especially from Pencil pine (MCK and CM) and Silver pine (AHA) although strong beta weights are also noted for King Billy pine (MDR), Huon pine (MHP) and Pink pine (DPP). Other parameters that provide useful information in the modelling are RW (King Billy pine (MWWTRL) and Huon pine (MHP)), LWB (Pencil pine (MCK, CM), Huon pine (BUT) and Kauri (PKL)) and DB (Huon pine (MHP) and Pink pine DPP)). These results are consistent with the correlation response function analysis (Table 2), but it must be emphasised that the results shown in Table 3 are related to specific seasons (JF for Tasmania and DJ for New Zealand) and may not reflect the optimal season for individual species or parameters (Figure 3).

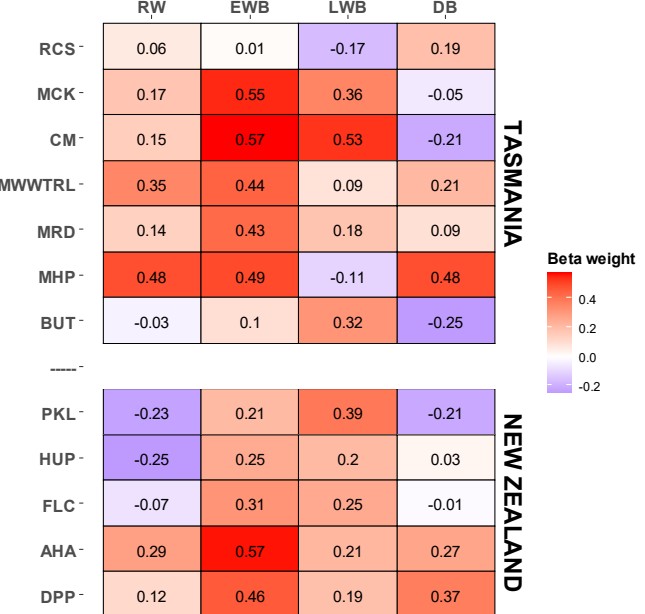

|          | RW    | EWB  | LWB   | DB    |
|----------|-------|------|-------|-------|
| RCS      | 0.06  | 0.01 | -0.17 | 0.19  |
| MCK      | 0.17  | 0.55 | 0.36  | -0.05 |
| CM       | 0.15  | 0.57 | 0.53  | -0.21 |
| MWWTRL   | 0.35  | 0.44 | 0.09  | 0.21  |
| MRD      | 0.14  | 0.43 | 0.18  | 0.09  |
| MHP      | 0.48  | 0.49 | -0.11 | 0.48  |
| BUT      | -0.03 | 0.1  | 0.32  | -0.25 |
| -----    |       |      |       |       |
| PKL      | -0.23 | 0.21 | 0.39  | -0.21 |
| HUP      | -0.25 | 0.25 | 0.2   | 0.03  |
| FLC      | -0.07 | 0.31 | 0.25  | -0.01 |
| AHA      | 0.29  | 0.57 | 0.21  | 0.27  |
| DPP      | 0.12  | 0.46 | 0.19  | 0.37  |

**Table 3: PC regression calibration (1901-1995) beta weights using all parameter and species data. The Tasmanian modelling was performed against January-February temperatures while New Zealand was with December-January.**

For the final countrywide calibration and validation experiments, three PC regression approaches were used, each reflecting more stringent screening procedures; (1) as already detailed above - all data entered into PCA and PCs with an eigenvalue > 1.0 that correlated significantly (95%) with the instrumental target were entered as possible candidates into a stepwise multiple regression; (2) same as (1) but chronologies were initially screened for significant correlation with the full period instrumental target before PCA; (3) similar to previous variants, but significant consistent correlations between the chronologies and the instrumental target for both the 1901-1950 and 1951-1995 periods were required.

For Tasmania, the initial 28 parameter chronologies were reduced to 17 and 10 respectively via the two more stringent screening procedures while the 20 initial chronologies from New Zealand were reduced to 13 and 7 respectively (Figure 4). Full period (1901-1995) calibration is excellent for all versions with the Tasmanian variants 1 and 2 expressing 63-64% of the JF temperature variance, reducing to 55% for variant 3. The New Zealand data return similarly good results with 61-64% of the DJ temperature variance being explained by all variants. Split period calibration and validation are equally good for all variants with the Tasmanian variants explaining 52-65% of the variance for all early/late period calibration while CE ranges from 0.49-0.61. Similar results are obtained for New Zealand with calibration adjusted $r^2$ ($_a r^2$) and CE values ranging from

0.53-0.78 and 0.43-0.77 respectively. For both countries, calibration and validation are marginally stronger for the later
1951-1995 period which might suggest some degree of uncertainty in the instrumental period in the early part of the 20th
century.

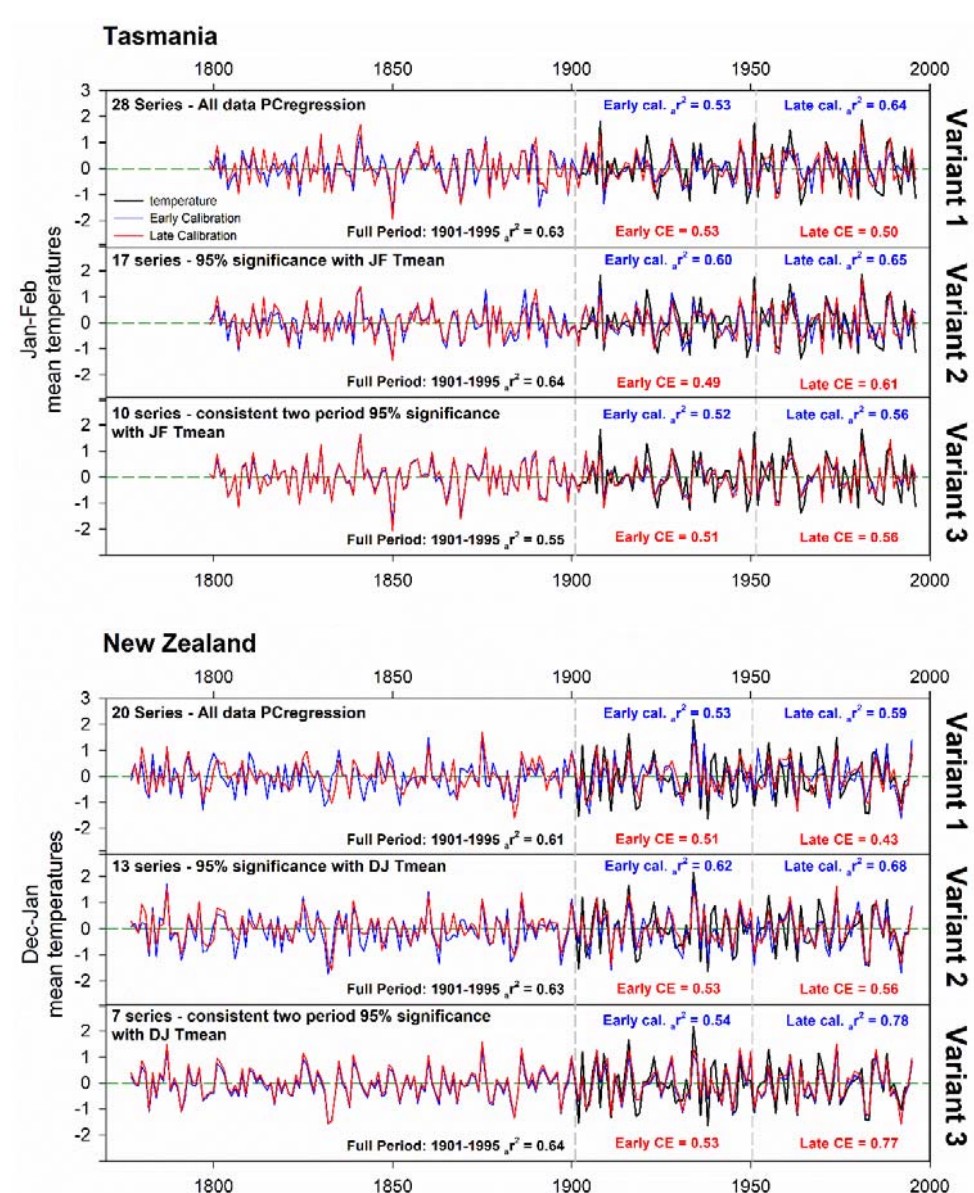

**Figure 4: Principal component regression results using all data (Variant 1), full period (1901-1995) screened data (Variant 2), and**
**two-period screening (Variant 3). Split period calibration and validation were performed over 1901-1950 and 1951-1995.**

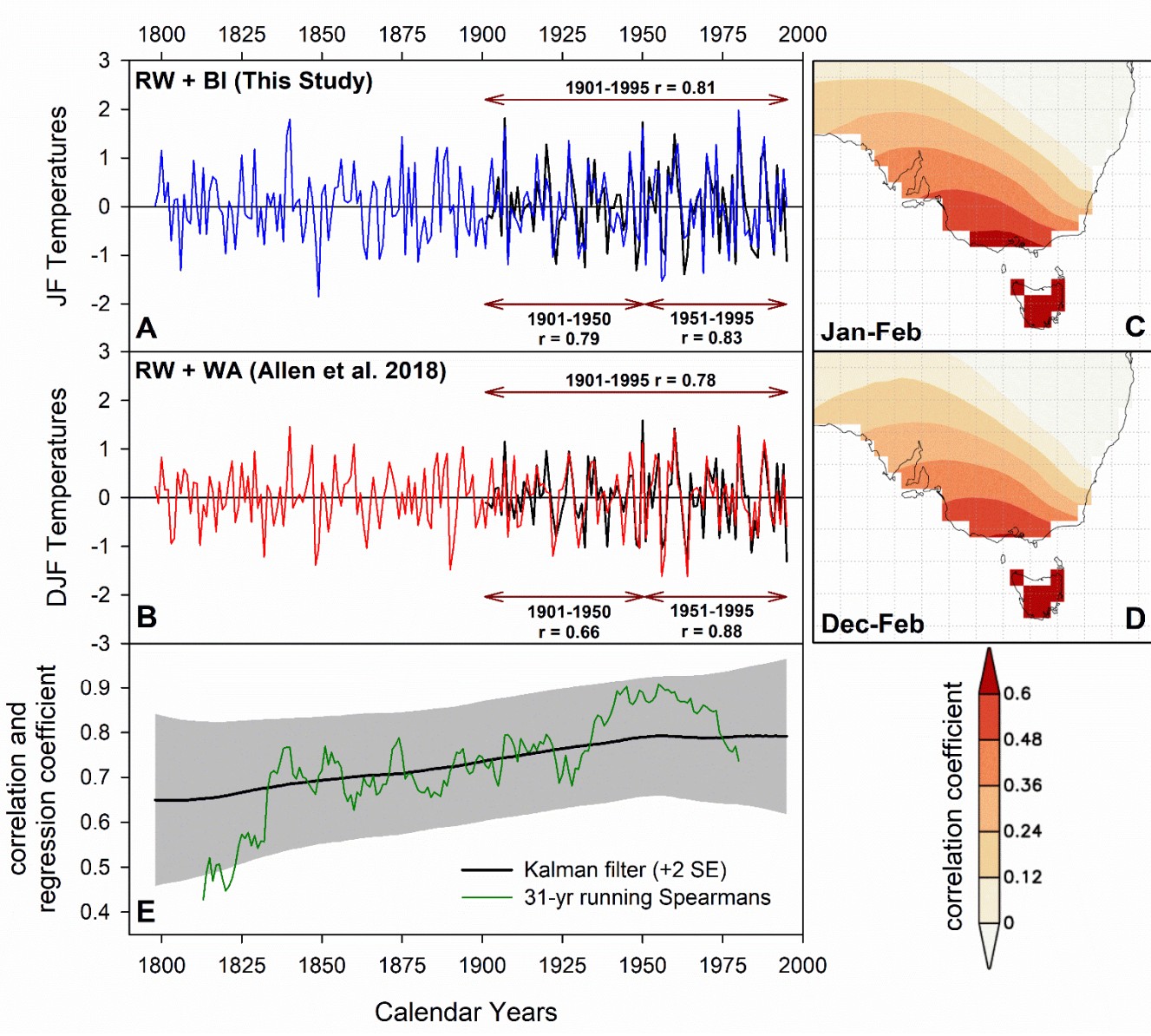



**Figure 5: A: Variant 2 (full period screened) Tasmanian JF temperature RW+BI parameter-based reconstruction with CRU TS**
**temperature data. Pearson's correlation is shown for 1901-1995, 1901-1950 and 1951-1995 period; B: As A, but for Allen et al.**
**(2018) RW+WA based Tasmanian DJF temperature reconstruction. These data have also been high-pass filtered using a 20-year**
**cubic smoothing spline; C + D: Spatial correlations (1841-1995) between each reconstruction and similarly detrended Berkeley**
**gridded data for the Jan-Feb and Dec-Feb seasons respectively; E: Running 31-year Spearman's rank correlation and Kalman**
**filter analysis (Visser and Molenaar 1988).**

Overall, the temperature reconstruction experiments for both Tasmania and New Zealand (Figure 4) return excellent results
with overall calibration $_ar^2$ values well above 0.60. Although no QWA data exist yet for New Zealand, Allen et al. (2018)
recently produced a range of PC regression-based Tasmanian summer temperature reconstructions from a network of 58
chronologies using RW and QWA (mean tracheid radial diameter, mean cell wall thickness, mean density and microfibril
angle). These variables were measured using the SilviScan system (Evans, 1994) from the same four Tasmanian tree species
used herein using samples from the same region, but from more sites. Strong calibration results explaining 50-60% of the
temperature variance and robust validation were also noted in their analyses. We compare our full period screened
temperature reconstruction (Variant 2, Figure 4 – representing the most used data screening approach in dendroclimatology)
with a high-pass filtered (20-year spline) version of Allen et al.'s (2018) "Berkeley all-data" reconstruction variant (Figure
5). Both reconstructions correlate similarly with the CRU TS temperature data (1901-1995: RW/BI r = 0.81 (JF) and
RW/WA r = 0.78 (DJF) – Figure 5a/b) although the BI-based reconstruction expresses a slightly more stable response with
temperatures over the 1901-1950 and 1951-1995 periods (r = 0.79 and 0.83 vs 0.66 vs 0.88). However, as the BI-based
reconstruction was calibrated against these CRU TS data, this slight difference may simply reflect the optimised PC
regression fit to one instrumental dataset over another. Equivalent split period correlations using the Berkeley temperature
data (Rohde et al. 2013), as used by Allen et al. (2018), are 0.82/0.86 (RW/BI) and 0.77-0.90 (RW/WA).

Correlation with the Berkeley data over the 1841-1900 period, shows that coherence is weaker but similar between Allen et
al.'s (2018) and this study (0.54 and 0.66). The spatial representation of the reconstructed temperature signal in both datasets
is almost identical when using linearly detrended Berkeley gridded temperature data (Rohde et al. 2013) even when
including data back to 1841 (Figure 5c/d). Both reconstructions are strongly correlated with each other (Pearson's r = 0.75,
1798-1995) although this coherence weakens back in time as evidenced by both a running 31-year Spearman's rank
correlation and Kalman filter (Visser and Molenaar 1988), showing a peak coherence in the 20th century that decreases back
towards the early part of the 19th century (Figure 5e). This likely represents the decrease in sample replication through time
in some BI-based datasets (MWWTRL and BUT) used in this study (Figure A1). Overall, the BI data, at least for Tasmania,
basically express the same high-frequency signal as the WA data used in Allen et al. (2018) and the results herein suggest
that BI parameters could provide excellent proxies of past growing season temperatures. However, for their potential to be
truly realised, the heartwood/sapwood colour change and other discolouration issues need to be overcome.
**4 Conclusions and future research directions**
In this study, we measured a range of blue intensity parameters from eight conifer species from Tasmania and New Zealand
to ascertain whether the use of EWB, LWB and/or DB can improve upon previous RW-only based dendroclimatic
reconstructions that explain about 40-45% of the temperature variance. No attempt to remove resins was made for this proof-
of-concept study. Therefore, due to the impact on intensity-based parameters of resins and heartwood/sapwood colour

changes on the wood, we detrended the chronologies and climate data using a very flexible spline (20-years) to focus only on the high-frequency signal. Metrics denoting signal strength (RBAR and EPS) indicated a very weak common signal in the BI parameters (mean RBAR range 0.14 – 0.16, Figure 2b) compared to the RW data (mean RBAR = 0.33) which appeared to be partly related to the relative variance in these datasets. The EWB data in particular exhibit very low variability which may mean that any colour variation in the wood that does not reflect true year-to-year wood anatomical variance may have a large impact on such data, thus weakening the common signal.

Despite the weak common signal expressed by the BI parameters, the climate signal extant in these data is very strong, especially EWB. When all parameters are combined using PC regression, depending on the period used, 52-78% of the summer temperature variance can be explained (Figure 4). This is generally greater than the norm for Northern Hemisphere based MXD/BI-related temperature reconstructions (Wilson et al. 2016), although admittedly, the results in this study are focused only on the high-frequency fraction of the data. These strong calibration results are driven mainly by EWB data from Pencil pine, high elevation Huon pine and King Billy pine (Tasmania) and Silver pine, Pink pine and cedar (New Zealand) although useful information was also identified in LWB (Pencil pine, low elevation Huon pine, Kauri and cedar), DB (high elevation Huon pine and Pink pine) and RW (high elevation Huon pine – Table 3). However, the relationship of LWB for most species with summer temperatures is opposite to that observed in the Northern Hemisphere and further study is needed to assess the physiological processes leading to this inverse relationship in these particular Southern Hemisphere conifers.

The similarity of the Tasmanian multi-TR-proxy reconstruction with a reconstruction heavily dependent on QWA data (Allen et al. (2018) - Figure 5) clearly highlights that the BI and WA data express similar wood properties. This is a highly encouraging result for the utilisation of BI as it is quicker and cheaper to produce than QWA data. However, the "elephant in the room" is whether robust low-frequency information can be extracted from BI-based parameters or is it an analytical methodology that ultimately will be relevant only for decadal and higher frequencies. It is unlikely that the heartwood/sapwood colour change (both sharp and gradual – Figure A1), expressed by most of the tree species used in this study, can be fully removed by resin extraction alone. Some success at overcoming heartwood/sapwood colour bias using DB has been shown for some Northern Hemisphere conifer species (Björklund et al., 2014a/b; Wilson et al., 2017b; Fuentes et al. 2017; Reid and Wilson 2020), but the DB results detailed herein (Table 3, Figures 2- 4) suggest that DB may not always provide a robust solution to the issue.

Other statistical approaches have been used to overcome the colour bias using either contrast adjustments (Björklund et al., 2014b; Fuentes et al., 2017) or band-pass approaches where the low-frequency signal is derived from the RW data and the high frequency is driven by the BI data (Rydval et al., 2017) but further experimentation is needed. We hypothesise that relatively sharp changes in colour intensity measures related to the heartwood/sapwood transition can be viewed

conceptually in a similar way to how endogenous disturbances affect ring-width parameters over time (Cook 1987). Similar
to the progress in developing growth release detection methods to reconstruct canopy disturbance histories of forests
(Altman 2020, Trotsiuk et al. 2018), radial growth averaging (Lorimer and Frelich 1989) or time series methods
(Druckenbrod et al., 2013; Rydval et al. 2015) could be used to identify and remove the colour bias signature resulting from
the change in physiology from heartwood to sapwood. However, to facilitate such signal processing methods, more studies
are needed to directly compare both MXD and QWA data with BI parameters to understand the secular trend biases in these
light intensity parameters. At the very least, the results detailed herein, based on a limited number of sites per species, show
that BI parameters can be used to identify those species that should be targeted for more costly and time-consuming
analytical methods such as QWA measurement.
**5 Appendix**

| SITE code | Mean value | CV | RBAR | n-EPS (0.85) |
|---|---|---|---|---|
| TASMANIA | | | | |
| RCSrw | 0.72 | 0.17 | 0.39 | 9.0 |
| RCSewb | 1.02 | 0.02 | 0.07 | 70.2 |
| RCSlwb | 0.58 | 0.05 | 0.18 | 25.6 |
| RCSdb | 0.45 | 0.08 | 0.17 | 27.3 |
| MCKrw | 0.75 | 0.13 | 0.40 | 8.5 |
| MCKewb | 1.19 | 0.01 | 0.20 | 22.5 |
| MCKlwb | 0.78 | 0.03 | 0.25 | 17.4 |
| MCKdb | 0.40 | 0.05 | 0.16 | 30.6 |
| CMrw | 0.71 | 0.10 | 0.31 | 12.9 |
| CMewb | 1.22 | 0.01 | 0.26 | 16.0 |
| CMlwb | 0.84 | 0.03 | 0.29 | 14.0 |
| CMdb | 0.36 | 0.05 | 0.19 | 23.5 |
| MWWTRLrw | 0.59 | 0.12 | 0.29 | 14.0 |
| MWWTRLewb | 1.23 | 0.01 | 0.14 | 34.9 |
| MWWTRLlwb | 0.82 | 0.03 | 0.15 | 31.1 |
| MWWTRLdb | 0.33 | 0.06 | 0.18 | 25.7 |
| MRDrw | 0.60 | 0.14 | 0.38 | 9.1 |
| MRDewb | 1.28 | 0.01 | 0.13 | 37.9 |
| MRDlwb | 0.88 | 0.03 | 0.21 | 21.2 |
| MRDdb | 0.40 | 0.07 | 0.21 | 21.3 |
| MHPrw | 0.36 | 0.13 | 0.39 | 8.9 |
| MHPewb | 1.06 | 0.02 | 0.28 | 14.7 |
| MHPlwb | 0.82 | 0.03 | 0.25 | 16.8 |
| MHPdb | 0.24 | 0.13 | 0.34 | 10.8 |
| BUTrw | 0.99 | 0.09 | 0.10 | 50.8 |
| BUTewb | 1.18 | 0.02 | 0.11 | 44.1 |
| BUTlwb | 0.63 | 0.06 | 0.11 | 47.0 |
| BUTdb | 0.58 | 0.07 | 0.10 | 52.9 |
| NEW ZEALAND | | | | |
| PKLrw | 1.16 | 0.17 | 0.40 | 8.4 |
| PKLewb | 1.17 | 0.01 | 0.12 | 40.9 |
| PKLlwb | 0.83 | 0.02 | 0.07 | 73.7 |
| PKLdb | 0.34 | 0.05 | 0.10 | 52.6 |
| HUPrw | 1.39 | 0.15 | 0.27 | 15.4 |
| HUPewb | 1.03 | 0.01 | 0.04 | 126.7 |
| HUPlwb | 0.73 | 0.02 | 0.03 | 190.4 |
| HUPdb | 0.29 | 0.04 | 0.02 | 314.5 |
| FLCrw | 0.44 | 0.16 | 0.44 | 7.3 |
| FLCewb | 0.97 | 0.02 | 0.12 | 41.4 |
| FLClwb | 0.76 | 0.02 | 0.14 | 36.2 |
| FLCdb | 0.21 | 0.09 | 0.20 | 22.0 |
| AHArw | 0.49 | 0.09 | 0.30 | 13.0 |
| AHAewb | 0.07 | 0.02 | 0.13 | 38.8 |
| AHAlwb | 0.04 | 0.05 | 0.16 | 29.6 |
| AHAdb | 0.02 | 0.10 | 0.11 | 43.8 |
| DPPrw | 0.48 | 0.11 | 0.33 | 11.7 |
| DPPewb | 0.89 | 0.02 | 0.12 | 41.9 |
| DPPlwb | 0.69 | 0.02 | 0.08 | 65.9 |
| DPPdb | 0.21 | 0.07 | 0.07 | 73.9 |



**Table A1: Mean RW, EWB, LWB and DB values for the raw chronologies. Coefficient of variation (CV) and mean inter-series**
**correlation (RBAR) are calculated from the 20-year spline detrended chronologies. n-EPS reflects the number of series needed to**
**attain an EPS value of 0.85 related to the RBAR value (Wilson and Elling 2004).**

**TASMANIA**

**RCS - Celery Top**

| | RCSewb | RCSlwb | RCSdb |
|---|---|---|---|
| **RCSrw** | 0.03 | -0.64 | 0.67 |
| **RCSewb** | | 0.22 | 0.27 |
| **RCSlwb** | | | -0.82 |

**MCK - Pencil Pine**

| | MCKewb | MCKlwb | MCKdb |
|---|---|---|---|
| **MCKrw** | -0.10 | -0.41 | 0.45 |
| **MCKewb** | | 0.57 | -0.01 |
| **MCKlwb** | | | -0.79 |

**CM - Pencil Pine**

| | CMewb | CMlwb | CMdbl |
|---|---|---|---|
| **CMrwl** | 0.01 | -0.30 | 0.43 |
| **CMewb** | | 0.68 | -0.04 |
| **CMlwb** | | | -0.71 |

**MWWTRL - King Billy Pine**

| | MTewb | MTlwb | MTdb |
|---|---|---|---|
| **MTrw** | 0.31 | -0.40 | 0.62 |
| **MTewb** | | 0.21 | 0.40 |
| **MTlwb** | | | -0.76 |

**MRD - King Billy Pine**

| | MRDewb | MRDlwb | MRDdb |
|---|---|---|---|
| **MRDrw** | 0.18 | -0.61 | 0.65 |
| **MRDewb** | | 0.14 | 0.44 |
| **MRDlwb** | | | -0.78 |

**MHP - Huon Pine (high elevation)**

| | MHPewb | MHPlwb | MHPdb |
|---|---|---|---|
| **MHPrw** | 0.64 | -0.19 | 0.69 |
| **MHPewb** | | 0.16 | 0.67 |
| **MHPlwb** | | | -0.56 |

**BUT - Huon Pine (low elevation)**

| | BUTewb | BUTlwb | BUTdb |
|---|---|---|---|
| **BUTrw** | -0.20 | -0.49 | 0.31 |
| **BUTewb** | | 0.21 | 0.35 |
| **BUTlwb** | | | -0.72 |

**NEW ZEALAND**

**PKL - Kauri**

| | PKLewb | PKLlwb | PKLdb |
|---|---|---|---|
| **PKLrw** | 0.14 | -0.37 | 0.54 |
| **PKLewb** | | 0.44 | 0.42 |
| **PKLlwb** | | | -0.55 |

**HUP - Kauri**

| | HUPewb | HUPlwb | HUPdb |
|---|---|---|---|
| **HUPrw** | -0.02 | -0.23 | 0.30 |
| **HUPewb** | | 0.58 | 0.23 |
| **HUPlwb** | | | -0.51 |

**FLC - NZ Cedar**

| | FLCewb | FLClwb | FLCdb |
|---|---|---|---|
| **FLCrw** | 0.40 | -0.47 | 0.70 |
| **FLCewb** | | 0.12 | 0.58 |
| **FLClwb** | | | -0.66 |

**AHA - Silver Pine**

| | AHAewb | AHAlwb | AHAdb |
|---|---|---|---|
| **AHArw** | 0.12 | -0.29 | 0.38 |
| **AHAewb** | | 0.48 | 0.32 |
| **AHAlwb** | | | -0.58 |

**DPP - NZ Pink Pine**

| | DPPewb | DPPlwb | DPPdb |
|---|---|---|---|
| **DPPrw** | 0.17 | -0.25 | 0.51 |
| **DPPewb** | | 0.60 | 0.60 |
| **DPPlwb** | | | -0.20 |



**Table A2: Correlation matrices for each site between the four detrended TR parameter chronologies (1798-1995). Grey shading**
**denotes a significant correlation (95%).**

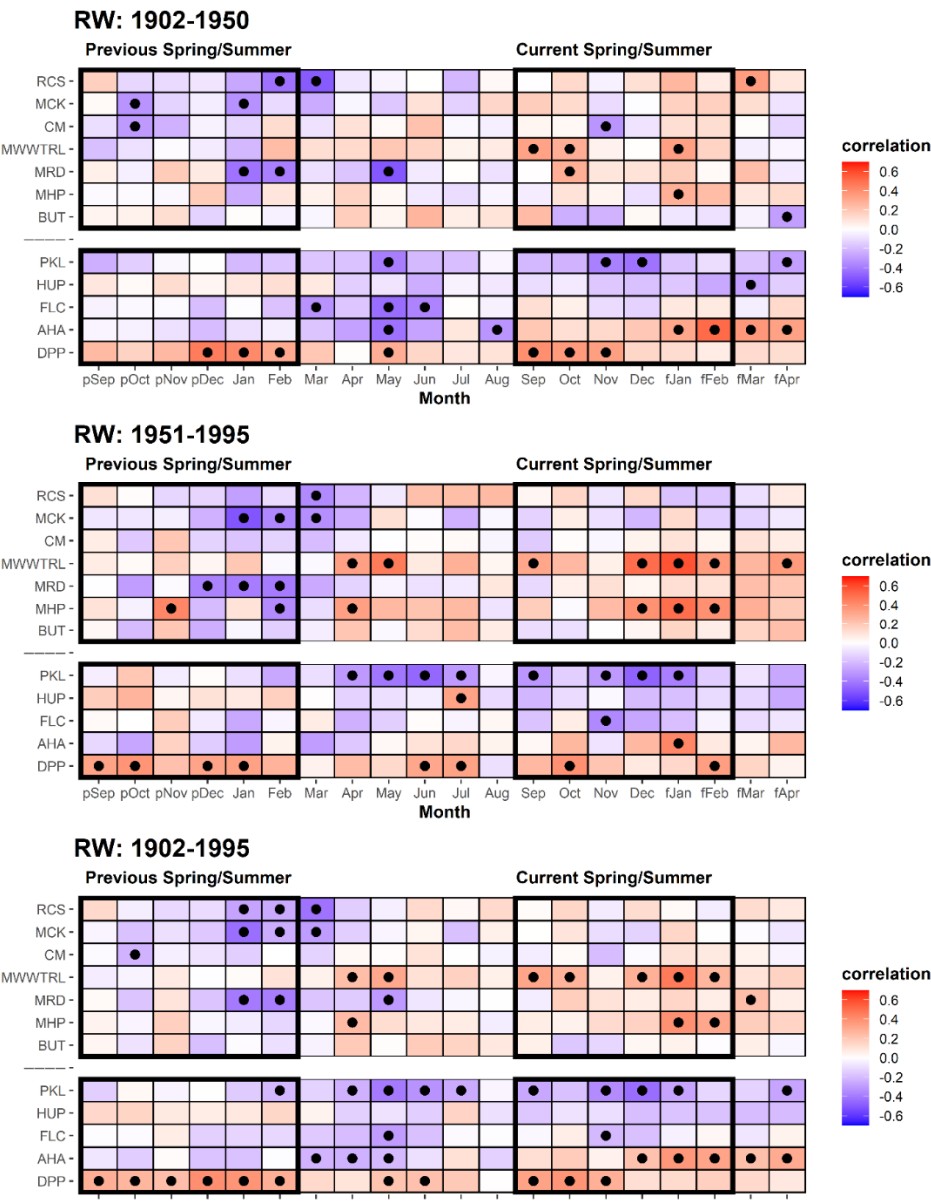



**Table A3a: Correlation response function analysis for ring-width with CRU TS temperatures. Analysis was undertaken over the**
**1902-1950, 1951-1995 and 1902-1995 periods. Black dots denoted correlations significant at the 95% C.L.**

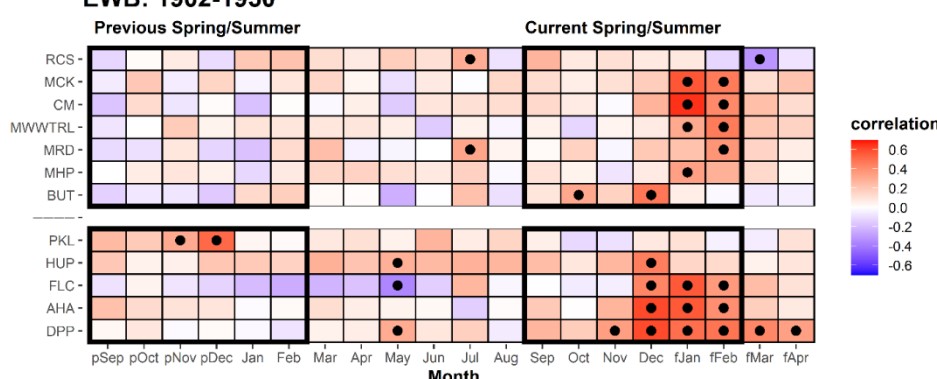

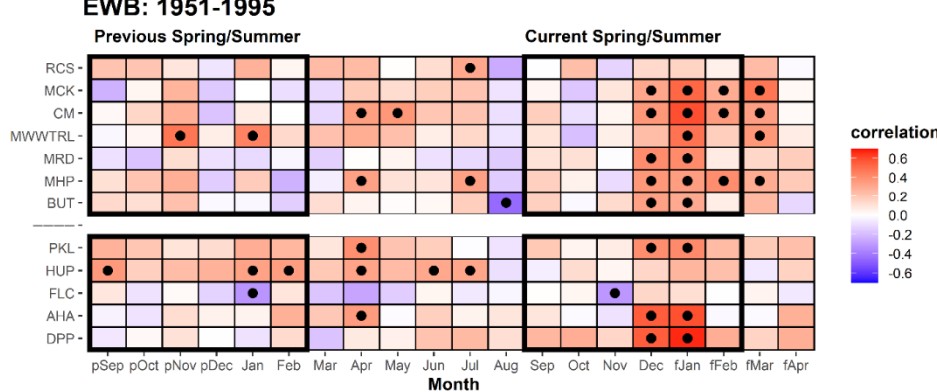

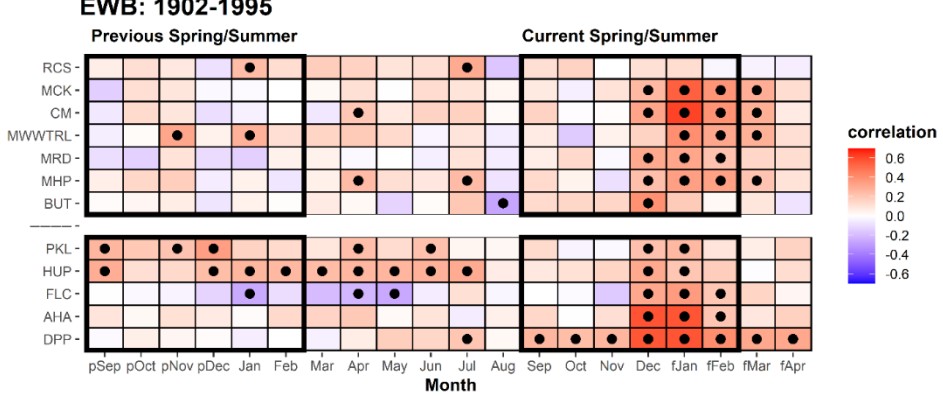


**Table A3b: As 3a but for EWB.**

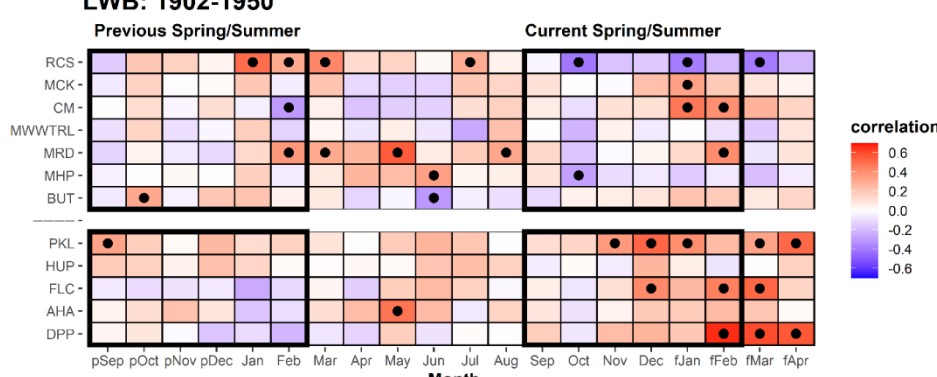

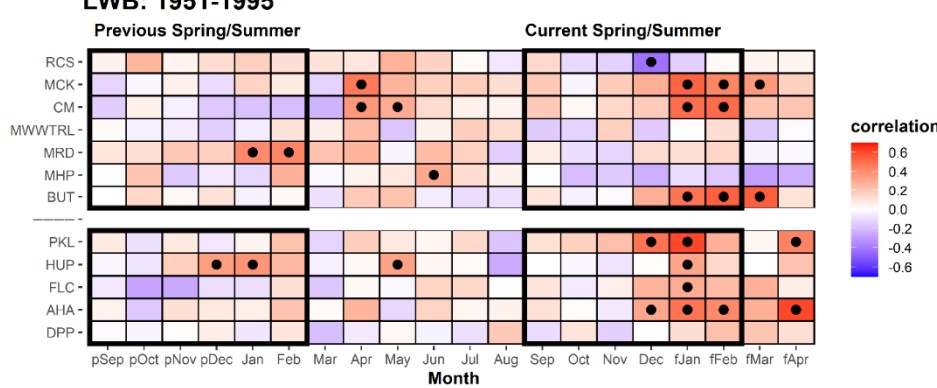

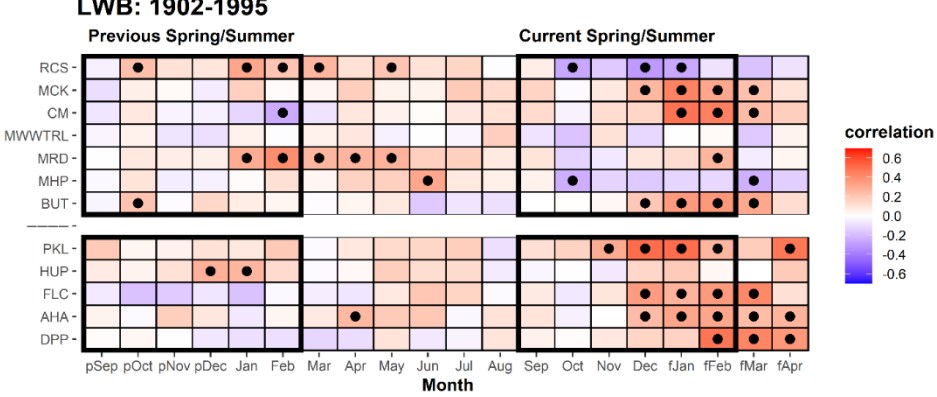


**Table A3c: As 3a but for LWB.**

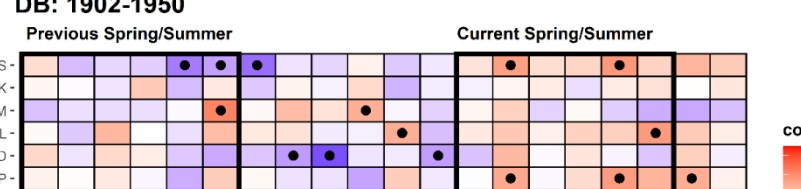

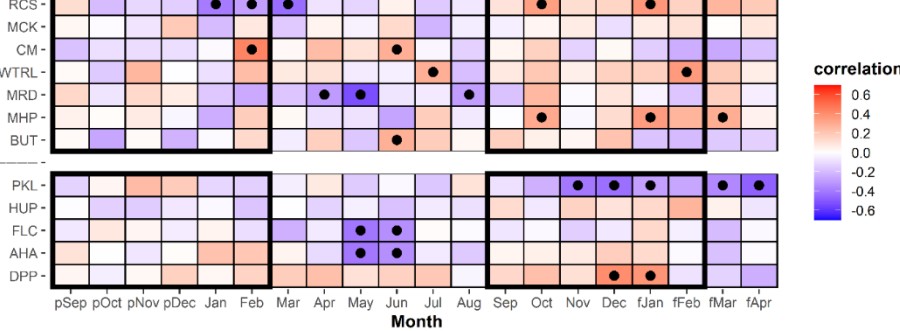

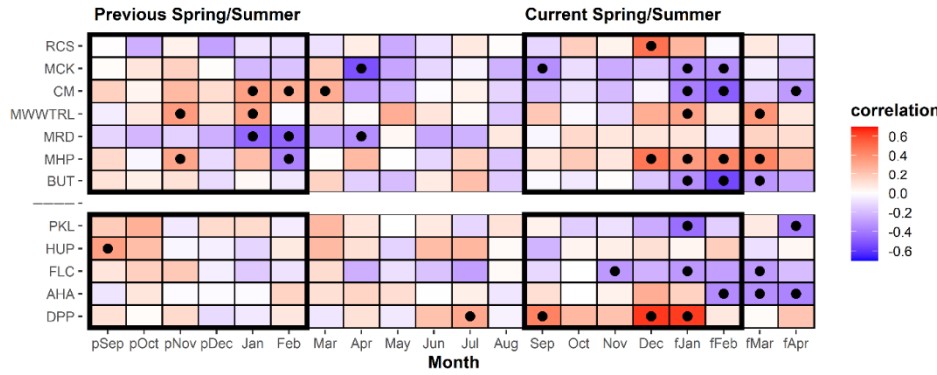

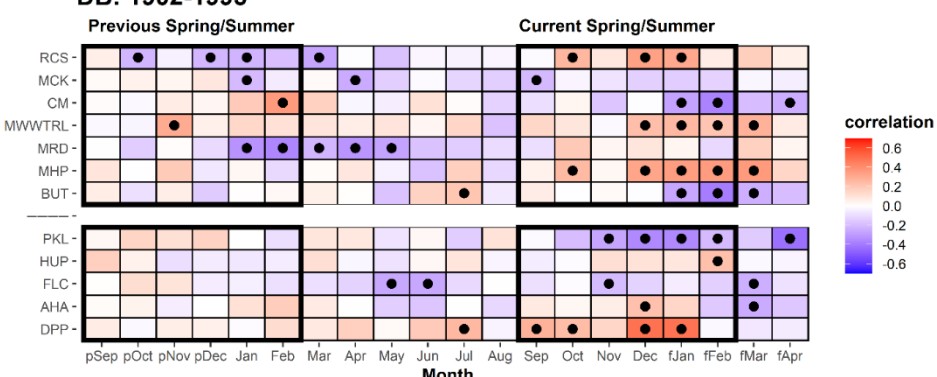


**Table A3d: As 3a but for DB.**


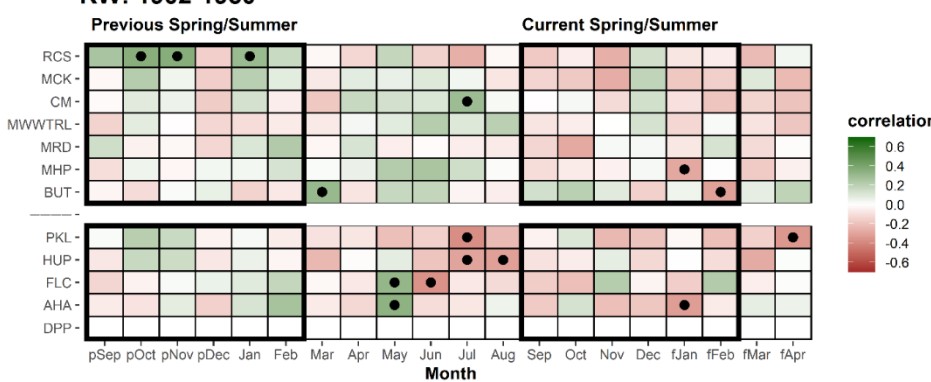

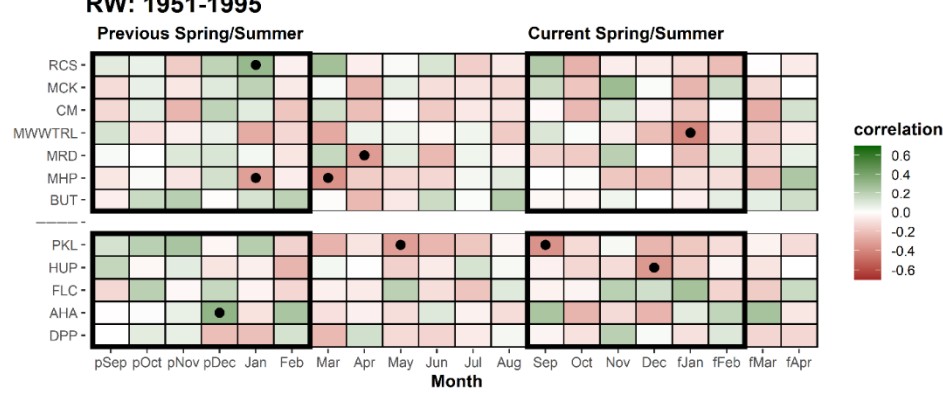

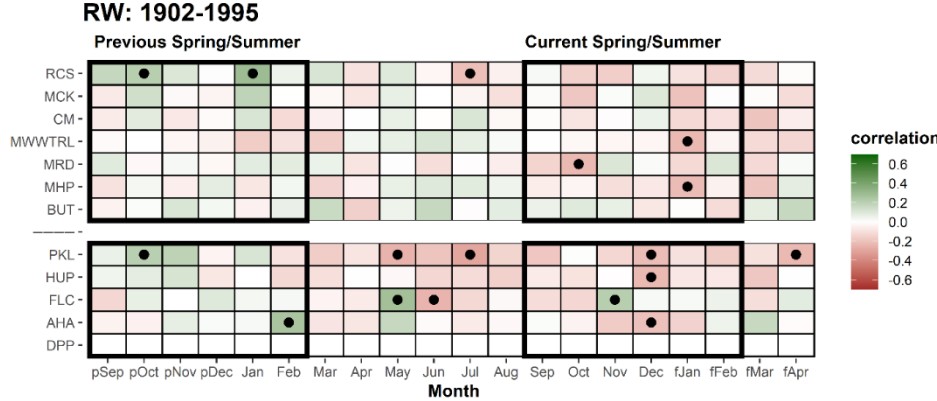


**Table A4a: Correlation response function analysis for ring-width with CRU TS precipitation. Analysis was undertaken over the**
**1902-1950, 1951-1995 and 1902-1995 periods. Black dots denoted correlations significant at the 95% C.L.**

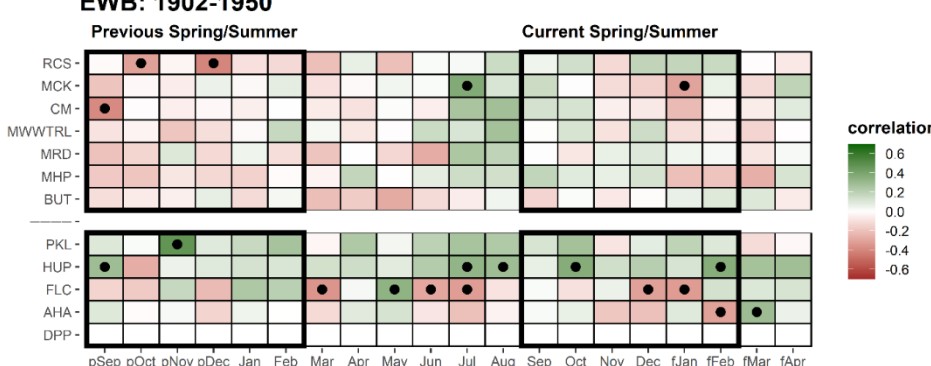

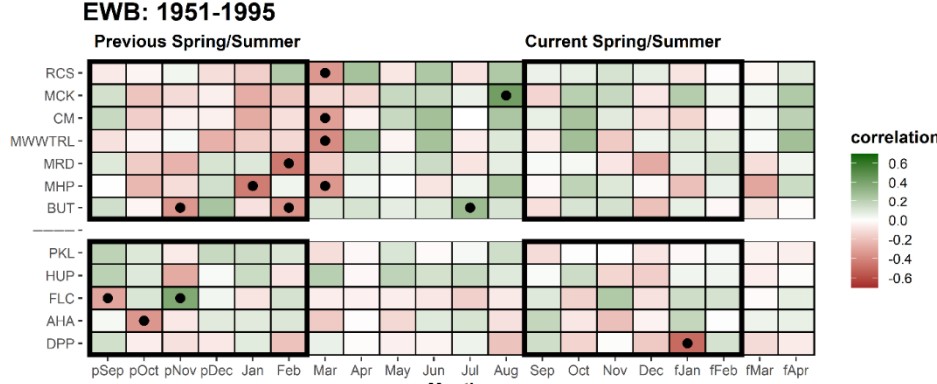

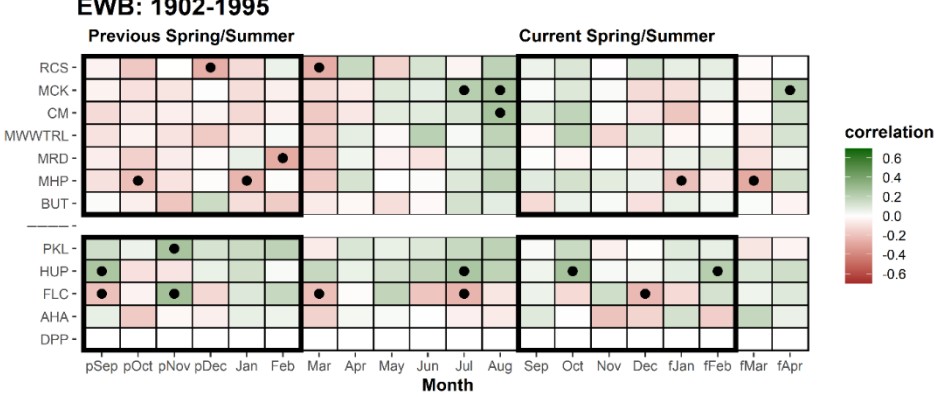


**Table A4b: As 4a but for EWB.**

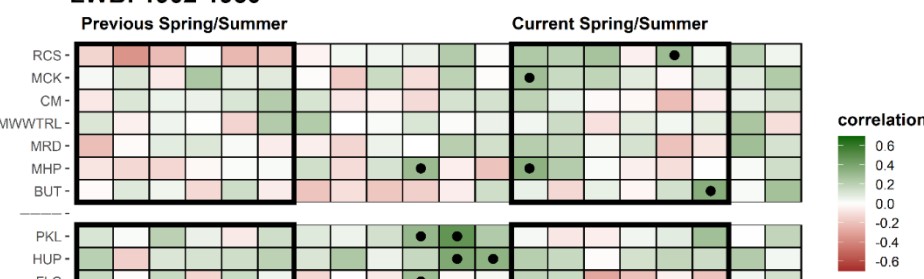

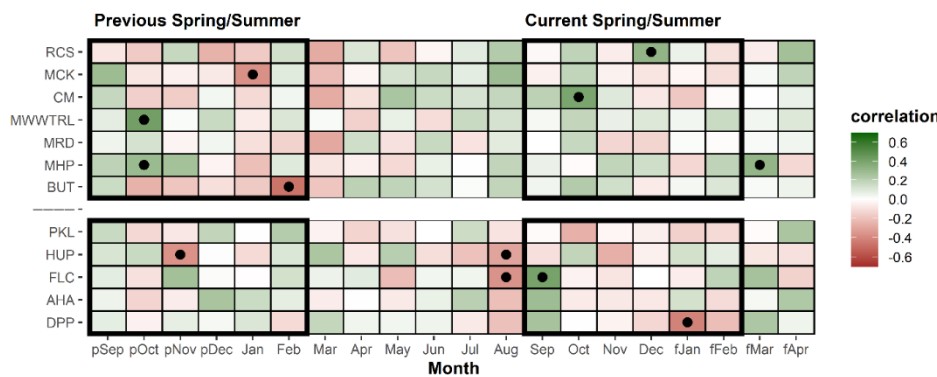

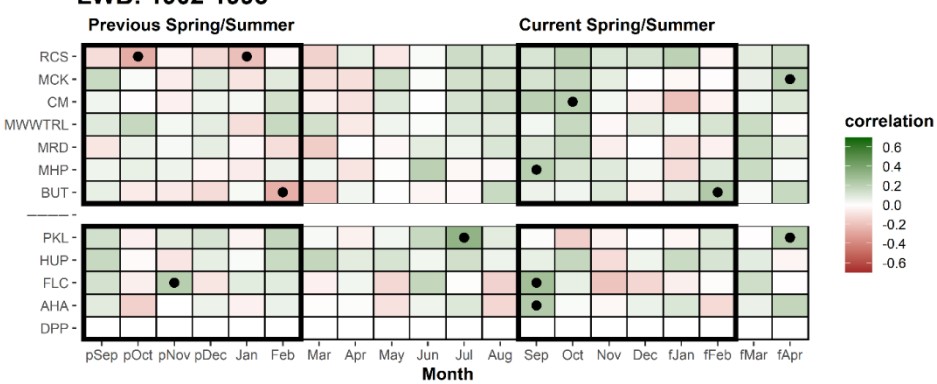


**Table A4c: As 4a but for LWB.**

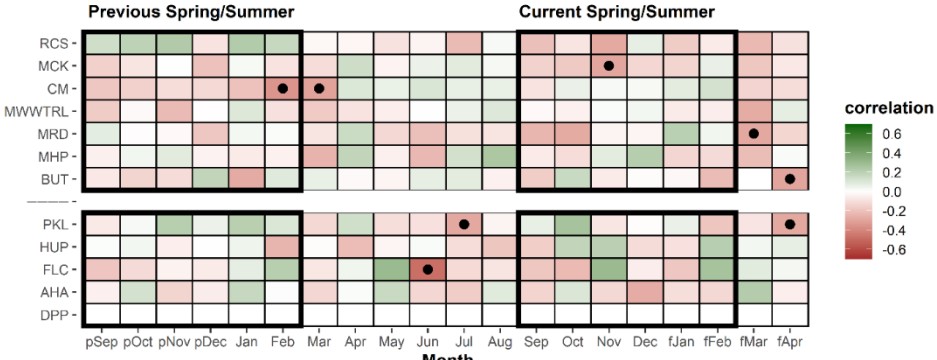

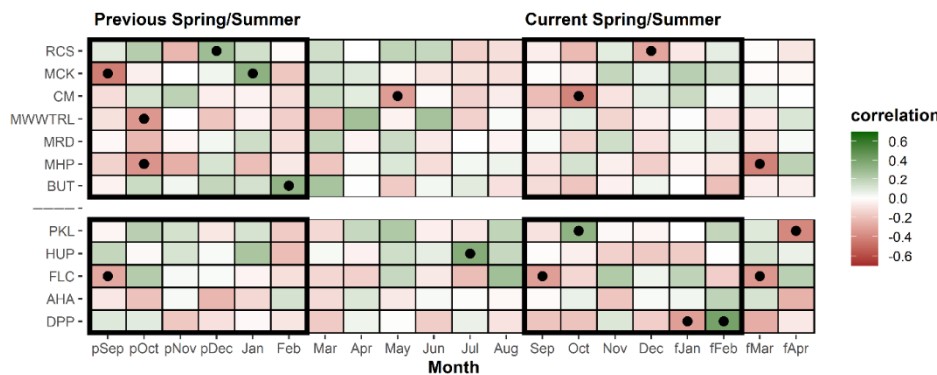

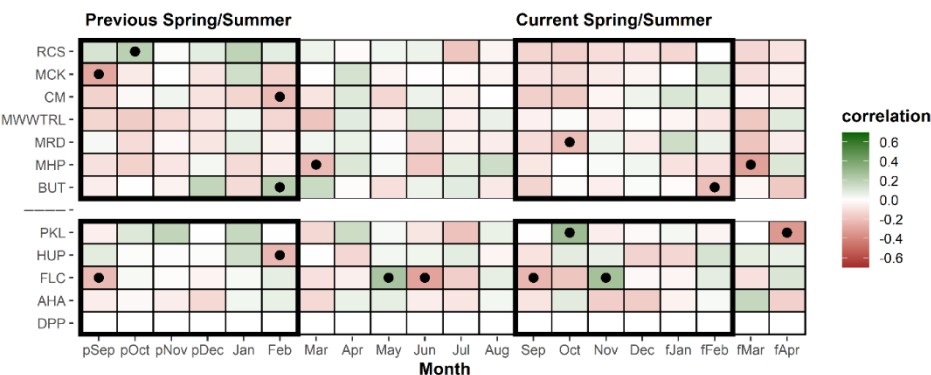


**Table A4d: As 4a but for DB.**

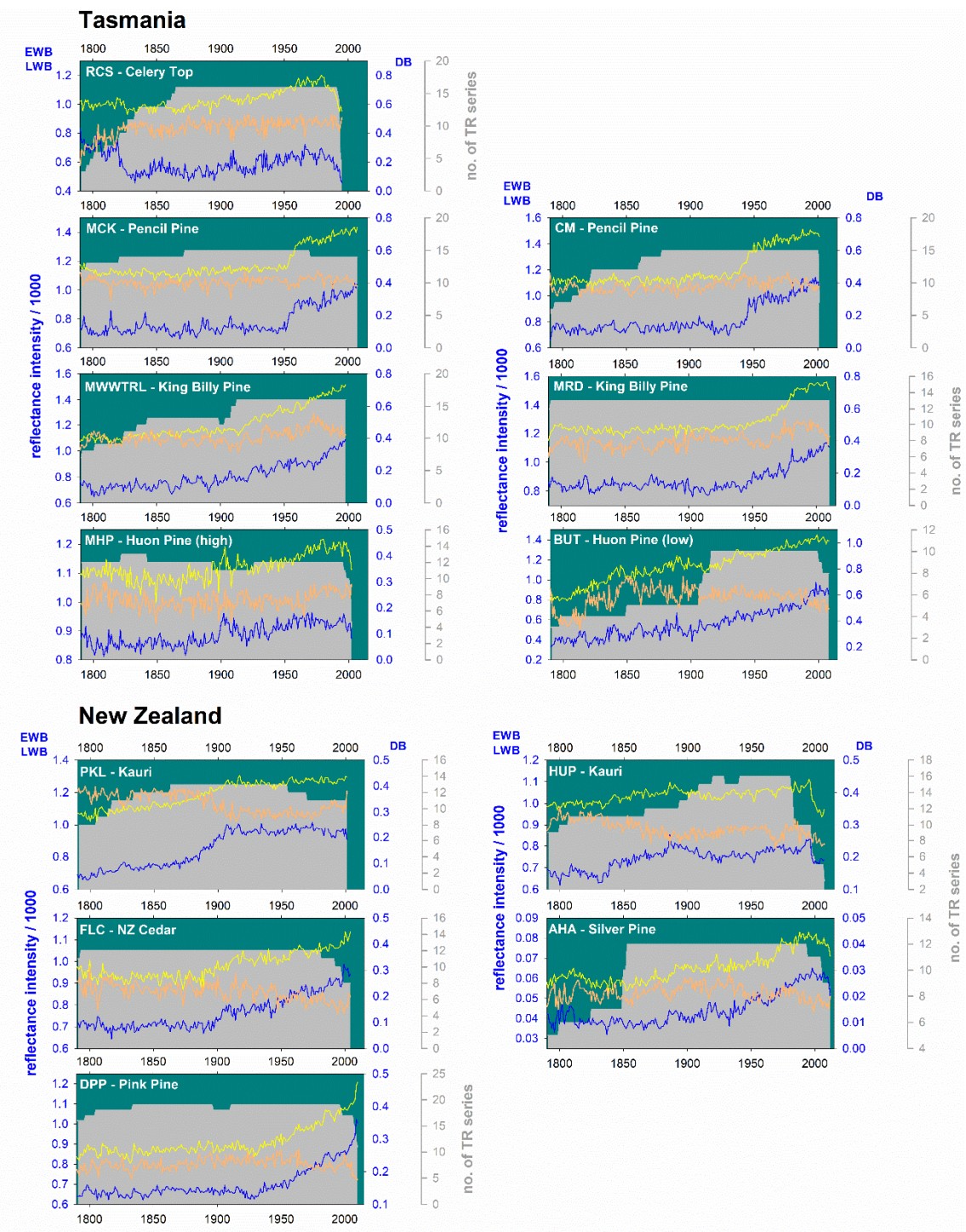


**Figure A1: Plots of raw mean chronologies of EWB (yellow), LWB (blue), DB (orange) and TR series replication (grey shading).**
**The left axis is for EWB and LWB, 1st right axis is DB, while 2nd right axis is series replication.**

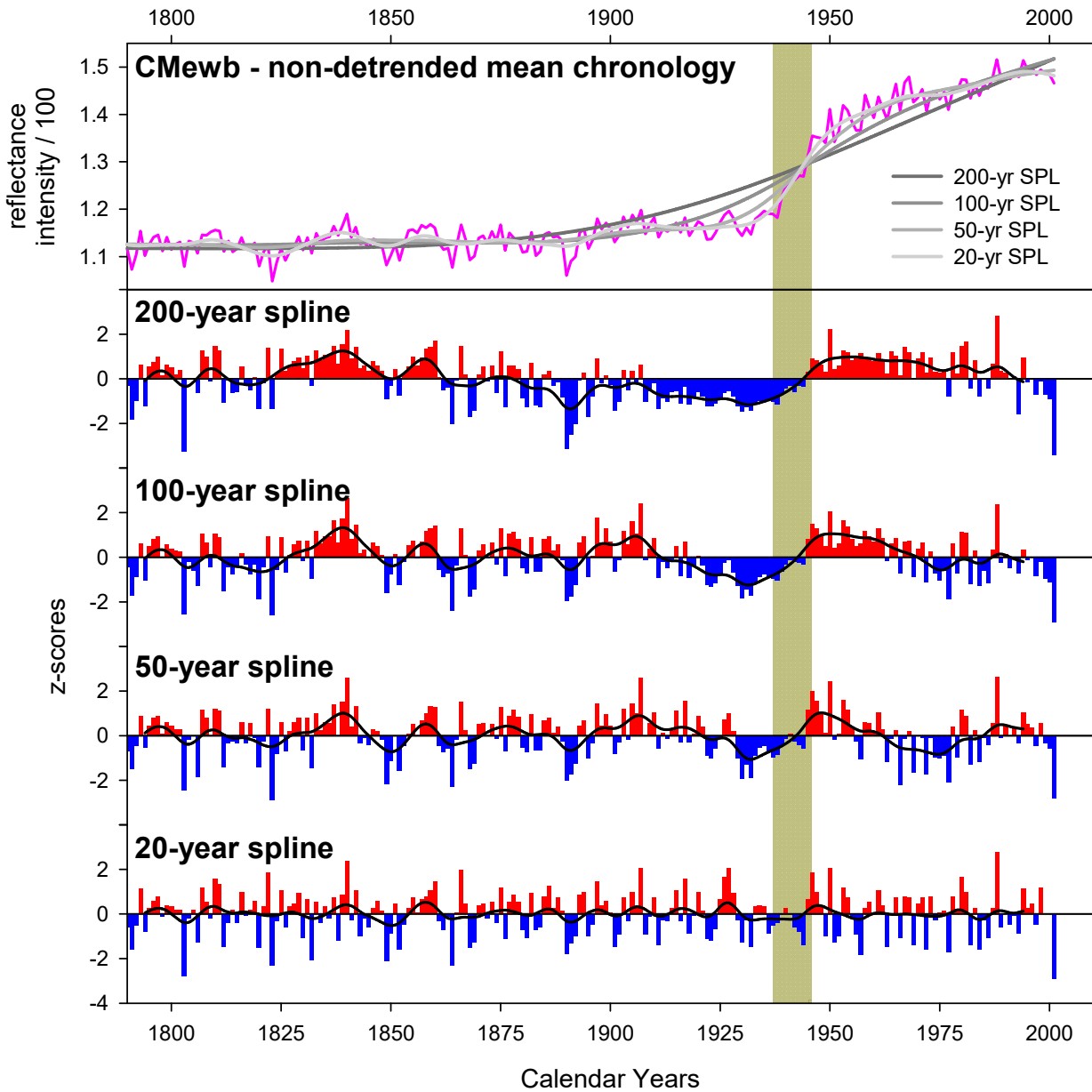

**Figure A2: Upper panel: raw mean non-detrended EWB chronology for the CM Pencil Pine site. lower panel: represents progressively more flexible spline detrending options. The vertical grey bar denotes the heartwood/sapwood transition period.**





## 6 Data availability

All raw data will be archived at the International Tree-Ring Databank on acceptance of the manuscript


## 7 Author contribution

RW: Project conception

RW, KA, PB, SB, GB, BB, EC, RD, AF, PK, JP: Sample collection and image acquisition

RW, KA, SB, MG: Data generation

All: Paper writing, final methodological design, comment and editing

## 8 Competing interests

The authors declare that they have no conflict of interest

## 9 Acknowledgements

RW was funded through the University of Melbourne Dyason Fellowship in 2014 to undertake preliminary measurement and analyses for this study. We also acknowledge NSF-NERC funding (NE/W007223/1). Permission to obtain samples from the Tasmanian sites was provided by Parks and Wildlife Tasmania through several different permits over multiple years. KA was supported by the Australian Research Council grants DP1201040320 and LP12020811 to PB. Permissions to obtain samples from New Zealand and Tasmanian sites were provided over multiple years by the New Zealand Department of Conservation and Parks and Wildlife Tasmania.

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
