# Peer review of "Evaluating the dendroclimatological potential of blue intensity on multiple conifer species from Tasmania and New Zealand"

_Biogeosciences, 2021_

## Author Comment (AC1)

**Please find below our responses to the Reviewer's comments. We have in fact made changes to the paper based on the reviewer's comments but hope we have communicated these changes in a clear way below.**
**Your feedback was very much appreciated and we hope will make the paper clearer.**

RC1:
The overall impression is very positive. The manuscript presents ambitious project of assessing usefulness Blue Intensity as climate proxy for Australia region. Although several interesting and valuable dendroclimatic studies exist for that part of world, a lot of though-provoking questions remain without answers. Substantial field, lab and intellectual effort back the manuscript. Needless to say the exploratory character of the research sometimes results in some incompleteness and shortcomings. It is untestable since the research is setting up the new directions of tree-ring studies in the region, not summing up some long persuaded topics. One of the shortcomings his is very much visible in way how the methods and results are handled.
*We hope the modified version of the paper is now clearer.*

It is difficult to follow the terminology, authors are joggling different BI parameters without proper descriptions, and what is worse without reasoning why they decided to use particular one instead of other. All these are rushed and create impression of rather chaotic structure. I strongly recommend putting some effort to make the manuscript clear and consistent it this aspect. Most of the elements are there therefore and it would be easy to improve the manuscript in terms of fluent presentation of research.
*We hope the modified version of the paper is now clearer.*
*Specifically – in the initial introduction we describe the general use of the term Blue Intensity (BI) in the current literature. We later refine the parameter definitions for this paper with the sentences: "To minimise nomenclature confusion, we refer to the reflectance parameters as earlywood blue intensity (EWB) and latewood blue intensity (LWB). Based on ecophysiological theory (Buckley et al. 2018) we posit that EWB, derived from maximum intensity values of the whole-ring reflectance spectrum, essentially provides a surrogate for mean lumen size of the earlywood cells, while LWB, derived from minimum reflectance values, reflects the relative density (i.e. proportion of cell wall to lumen area) of the darker latewood cell walls. " and later w.r.t. DB: "Björklund et al. (2014) proposed a statistical procedure that could correct for such colour changes. This procedure subtracts the LWB reflectance value from the EWB data producing a delta parameter (hereafter referred to as Delta BI - DB). Theoretically, DB should correct for common colour change biases between heartwood and sapwood and even resinous zones within the wood. To date, DB has been utilised successfully in only a few studies (Björklund et al., 2014, 2015; Wilson et al., 2017b; Fuentes et al. 2018; Blake et al. 2020; Reid and Wilson 2020)."*

Again the really missing pieces of information is how and why certain BI parameter were selected, e.g. it is first time that earlywood maximum blue reflectance intensity, latewood minimum blue reflectance intensity were employed in dendroclimatic analyses thus authors are obligated to provide the proper description of the parameter, especially what Blue Reflectance registers and what are the possible errors while measuring it. The publication of Buckley et al. 2018 is very good example how the introduction of the relatively new parameter can and should be done.
*This is not the first time that these parameters have been used in dendroclimatic analysis and Buckley is one of a growing number of papers. We hope now the methodology and overall description is clearer. We believe the parameter description is now clearer.*

The additional recommendation would be to describe vegetation and climate of the studied region in comprehensive way. The tree species, forest, and climate are exotic to most of the readers. We would be able to appreciate the importance of results much better knowing basic information about subject of the

research – trees and forest of this part of New World. The traditionally included in papers chapter description of the study site could be seen as outdated and not relevant but sometimes is beneficial and discussed manuscript is good example of it.

*We understand the reviewer's comment here and in the "Data Methods" section have expanded the site description to better communicate the relative positions of these sites to theoretical tree-line (elevation or latitude). We further emphasise that the samples chosen for this BI assessment were sites that have already been used in previous dendroclimatic analyses and there is a wealth of information in earlier papers that does not need to be repeated here. Finally, we would like to emphasise that the BI parameter (and associated QWA data) are identifying new parameter/climate relationships that are quite different to what we would expected from RW alone.*

*The new text for this section is: "Four tree species from Tasmania and four from New Zealand were targeted for analysis (Figure 1, Table 1) representing conifer species that have not only been the focus of previous dendrochronological studies, but each has the potential to produce climate proxy records substantially greater than 1000 years in length. Until recently, RW data were used for most Australasian dendroclimatological studies, with calibration results never exceeding 40-45% explained variance. In Tasmania, the strongest calibration results for summer temperatures had been obtained using high elevation Huon pine (Lagarostrobos franklinii - Buckley et al. 1997; Cook et al. 2006) although some coherence was also found for Pencil pine (Athrotaxis cupressoides) and King Billy pine (Athrotaxis selaginoides - Allen et al. 2011; Allen et al. 2017). The study sites (Table 1) for Pencil pine (MCK and CM) and King Billy pine (MWWTRL and MRD) are located close to the upper timberline limit of these species and growth is expected to be controlled mostly by summer temperatures. Likewise, the high elevation Huon pine (MHP) site is also close to the upper treeline where summer temperature is the dominant response (Buckley et al. 1997). However, BUT is located at the lower end of the Huon pine elevational range within a riparian environment so temperature limitation is unlikely in a traditional sense. However, Drew et al (2013) identified strong summer temperature signals in latewood QWA data for this site. Celery Top (Phyllocladus aspleniifolius) RW data, however, express a complex non-linear relationship with climate along its species' elevational range and have not been used for dendroclimatic reconstruction (Allen et al. 2001). By contrast, summer temperature calibration experiments performed on measurement series of several wood anatomical properties (e.g. tracheid radial diameter, cell wall thickness and microfibril angle), as well as RW and ring density, from these same species, have shown substantial improvement over RW alone (Allen et al. 2018), although these QWA data have been more useful for hydroclimate reconstructions (Allen et al. 2015a/b). In New Zealand, RW-based summer temperature reconstructions have been developed from NZ Cedar (Libocedrus bidwillii - Palmer and Xiong 2004), Silver pine (Manoao colensoi - Cook et al. 2002, 2006) and Pink pine (Halocarpus biformis - D'Arrigo et al. 1996, Duncan et al. 2010) although ring density (Xiong et al. 1998 – Pink pine) and BI (Blake et al. 2020 – Silver pine) measured from the earlywood have produced stronger results. For this study, we specifically measured BI from samples used in previous, mostly RW-based dendroclimatic, studies where summer temperature was found to be the dominant climate signal - at least for NZ Cedar, Silver pine and Pink pine. The sites for these three New Zealand species are close to their southern (latitudinal) limits (especially the Stewart Island Pink pine site) which is thought to compensate, to some degree, for their modest elevational range (Table 1). Kauri (Agathis australis) is the longest-lived tree species in Australasia (Boswijk et al. 2014) but only a few sites of reasonably mature trees exist. Previous analyses have identified a complex mixed response to both temperature and precipitation through the growing season (Buckley et al 2000, Fowler et al, 2000). However, it is notable that Kauri RW data express a strong stable relationship with indices of the El Nino Southern Oscillation (Cook et al. 2006; Fowler et al. 2012)."*

The last but not least issue concerns the title. It is inadequate to entitle manuscript "Evaluating the dendroclimatological potential of blue intensity on multiple conifer species from Australasia" having

virtually zero data from main part of that region – the Australia continent. I suggest authors try to imagine their reaction to the publication "Evaluating the dendroclimatological potential of blue intensity on multiple conifer species from Italy" based on sites only from Sicily and Sardinia. Technically this is territory of Italy but it is kind of obvious that something is missing, isn't it. Thus the title should be corrected to adjust to real geographical coverage of the research.

***The title is now changed to: "Evaluating the dendroclimatological potential of blue intensity on multiple conifer species from Tasmania and New Zealand"***

---

## Author Comment (AC2)

**Please find below our responses to the Reviewer's comments. We have in fact made changes to the paper based on the reviewer's comments but hope we have communicated these changes in a clear way below.**
**Your feedback was very much appreciated and we hope will make the paper clearer.**

RC2:

General comments:

The manuscript, which investigates the dendroclimatological potential of blue intensity parameter datasets from a broad range of Australasian conifers, represents an important addition to research into the potential utility of the underutilized blue intensity tree ring parameter in a dendroclimatological context. Overall, this study provides valuable information about the behavior, climate response, and reconstruction potential of blue intensity in parts of Australasia and helps open up a new frontier in the utilization of this climatically sensitive and affordable tree ring parameter in that region and the Southern Hemisphere more broadly. The manuscript is generally well written, nicely structured and logically organized, although some additional context or clarification would be helpful in certain parts of the text. However, there are several important aspects of the manuscript, which could be improved, addressed or more thoroughly discussed (detailed comments are provided in 'specific comments' below):

*We hope we have adequately addressed these issues below*

It is unfortunate that lower frequency trends were not assessed, as this would have greatly enhanced the significance of the study and the lack of resin extraction or other form of sample treatment represents a substantial limitation of this study. While this limitation is acknowledged and addressed by adopting a very flexible detrending approach with all analyses focused on high-frequency relationships, as a result, the conclusions that can be drawn are limited. For example, the full potential advantages of the DB dataset remain unexplored. With that in mind, interpretation of the results and some of the statements in the discussion could more explicitly acknowledge this limitation.

*The current preliminary study was exploratory in nature, and we purposely focussed on exploring the potential of BI parameters on species that had mostly used traditional ring-width (RW) for dendroclimatological studies. Due to the nature of the project, we purposely measured a relatively small number of samples allowing us the opportunity to test as many species as possible – a strategy we have earlier employed for Scotland, Pyrenees, Alaska and NW North America.*

*Therefore, at this time, replication is rather too low to robustly explore the low frequency problems which is a focus of ongoing work (we in fact now have a 3-year project just funded through NSF-NERC to specifically build on this preliminary work).*

*However, the reviewer is incorrect that we have not explored DB in any meaningful way. We would argue that if the DB parameter does not show any (or at least only a weak) high frequency climate fidelity, then this is a serious problem for the use of this parameter and it really does not matter what secular trends the DB data may or may not express.*

Considering the relatively high BI replication requirements to generate a representative chronology, it is not clear whether and to what extent some of the identified relationships could be affected by the relatively low and variable replication between sites and sub-optimal EPS.

*A high EPS value does not guarantee a strong climate signal (see Buras 2017, Dendrochronologia). However, the BI literature is rather consistent in showing that BI parameters often have strong climate relationships despite weaker signal strength than standard ring density parameters. The fact that the current results suggest strong climate signals despite the weak expressed signal strength simply suggests that the results will likely improve as replication is increased.*

*Again, we simply re-iterate that this study explored the potential of BI parameters for these select species and never aimed to produce a refined reconstruction. The good comparison with the QWA reconstruction from Tasmania, in our minds, provides very encouraging results - especially when the dataset is far from ideal.*

The quality of the gridded instrumental dataset with respect to its local representativeness over space and time in some locations could possibly affect the results. Also, the aggregation of the gridded data across New Zealand could potentially be problematic and should be examined in more detail.

**We do not agree that the gridded climate data needs to be examined in more detail. Earlier research by Allan et al (2019) has already explore the difference between station data and gridded products in Tasmania so it is not worth repeating here. We believe that the CRU TS data are perfectly adequate for the main aims of this paper and the focus on the high pass fraction also minimises potential heterogeneity problems (often expressed with different trends and amplitudes) between different stations. Low elevation stations will never be truly representative of high elevation tree ring sites.**

[Figure]

**W.r.t. New Zealand, we in fact already explored this issue in the paper and made a clear statement (line 225 in original submitted article) that the coherence between the southern NZ temperature data and the northern are strongly correlated with a mean inter-correlation between all season of 0.93 (STDEV = 0.01).**

**The map to the left shows the New Zealand met stations that extend back to at least 1901 (the start of the CRU TS data). The most northerly station (Auckland) correlates strongly with the most southerly (Dunedin) with a r value of 0.82 for December-January mean temperatures.**

[Figure]

**Correlations (common period: 1906-1970) and time-series (left) of the northern, southern CRU TS grids and Auckland and Dunedin DJ temperatures clearly show the strong coherence from north to south. These results suggest strongly that the gridded temperature products represent their respective regions well.**

Although the importance of site elevation / latitude in relation to the treeline is highlighted in the introduction, there appears to be no consideration of such factors in this study, which could certainly affect the RW results and may also play a significant role in influencing the climatic response of the BI parameters. Large differences in elevation between sites in Tasmania and New Zealand along with the large latitudinal range (and possible climate response gradient from north to south) of the New Zealand sites may also play an important role and should be considered.

*As stated for reviewer 1 - We understand the reviewer's comment here and in the "Data Methods" section have expanded the site description to better communicate the relative positions of these sites to theoretical tree-line (elevation or latitude). We further emphasise that the samples chosen for this BI assessment were sites that have already been used in previous dendroclimatic analyses and there is a wealth of information in earlier papers that does not need to be repeated here. Finally, we would like to emphasise that the BI parameter (and associated QWA data) are identifying new parameter/climate relationships that are quite different to what we would expected from RW alone.*

*The new text for this section is: "Four tree species from Tasmania and four from New Zealand were targeted for analysis (Figure 1, Table 1) representing conifer species that have not only been the focus of previous dendrochronological studies, but each has the potential to produce climate proxy records substantially greater than 1000 years in length. Until recently, RW data were used for most Australasian dendroclimatological studies, with calibration results never exceeding 40-45% explained variance. In Tasmania, the strongest calibration results for summer temperatures had been obtained using high elevation Huon pine (Lagarostrobos franklinii - Buckley et al. 1997; Cook et al. 2006) although some coherence was also found for Pencil pine (Athrotaxis cupressoides) and King Billy pine (Athrotaxis selaginoides - Allen et al. 2011; Allen et al. 2017). The study sites (Table 1) for Pencil pine (MCK and CM) and King Billy pine (MWWTRL and MRD) are located close to the upper timberline limit of these species and growth is expected to be controlled mostly by summer temperatures. Likewise, the high elevation Huon pine (MHP) site is also close to the upper treeline where summer temperature is the dominant response (Buckley et al. 1997). However, BUT is located at the lower end of the Huon pine elevational range within a riparian environment so temperature limitation is unlikely in a traditional sense. However, Drew et al (2013) identified strong summer temperature signals in latewood QWA data for this site. Celery Top (Phyllocladus aspleniifolius) RW data, however, express a complex non-linear relationship with climate along its species' elevational range and have not been used for dendroclimatic reconstruction (Allen et al. 2001). By contrast, summer temperature calibration experiments performed on measurement series of several wood anatomical properties (e.g. tracheid radial diameter, cell wall thickness and microfibril angle), as well as RW and ring density, from these same species, have shown substantial improvement over RW alone (Allen et al. 2018), although these QWA data have been more useful for hydroclimate reconstructions (Allen et al. 2015a/b). In New Zealand, RW-based summer temperature reconstructions have been developed from NZ Cedar (Libocedrus bidwillii - Palmer and Xiong 2004), Silver pine (Manoao colensoi - Cook et al. 2002, 2006) and Pink pine (Halocarpus biformis - D'Arrigo et al. 1996, Duncan et al. 2010) although ring density (Xiong et al. 1998 – Pink pine) and BI (Blake et al. 2020 – Silver pine) measured from the earlywood have produced stronger results. For this study, we specifically measured BI from samples used in previous, mostly RW-based dendroclimatic, studies where summer temperature was found to be the dominant climate signal - at least for NZ Cedar, Silver pine and Pink pine. The sites for these three New Zealand species are close to their southern (latitudinal) limits (especially the Stewart Island Pink pine site) which is thought to compensate, to some degree, for their modest elevational range (Table 1). Kauri (Agathis australis) is the longest-lived tree species in Australasia (Boswijk et al. 2014) but only a few sites of reasonably mature trees exist. Previous analyses have identified a complex mixed response to both temperature and precipitation through the growing season (Buckley et al 2000, Fowler et al, 2000). However, it is notable that Kauri RW data express a strong stable relationship with indices of the El Nino Southern Oscillation (Cook et al. 2006; Fowler et al. 2012)."*

While a consistent EWB response does genuinely appear to exist throughout this network, the broad and diverse range of relationships between all of the parameters across species and their responses suggests that (at least w.r.t. assessing climate response) it may be more appropriate to investigate and discuss such inter-connections in more detail at an individual site or species level. Combining all data together and

extracting the overriding signals using multivariate techniques may overlook other potentially useful climatic signals and relationships present in subsets of the full dataset, especially since multiple species and parameters are involved. This could be evaluated with a more nuanced approach.

*There are three levels of effective assessment of the BI data in the paper; (1) individual site and parameter assessment using correlation response function analysis with temperature and precipitation (original Figure 3 and appendix figures A3a – A4d); (2) parameter and species multivariate assessment (Figure 4) and (3) and the full multi-species/parameter analysis (Table 2 and Figure 5).*

*In our opinion we already provide a substantial assessment of each species and parameter from individual sites up to the full networks for each region.*

Although the BI-based dataset calibrates very well and is indeed comparable to the QWA-based reconstruction within the context of this particular study, some of the statements in the manuscript may somewhat exaggerate the capabilities of BI in relation to QWA even without considering the lower frequency limitations of BI.

*Our current statements reflect on the potential of the BI data in the context of utilising a multi-species/site data which was the same approach as used by Allen et al (2018). In the context of the high frequency signal, we believe the results are rather compelling. Please remember that the current datasets are far from perfect – no resin extraction, low replication, weak signal strength etc – yet, we can still extract a very robust climate signal at high frequencies. With multivariate ar2 values well in excess of 0.50, we feel strong statements of optimism are warranted. There is plenty of cautionary text about the challenges of extracting a meaningful low frequency signal, but there is a substantial number of successful studies from the northern hemisphere now, why could we not think that it would not be possible here in the future with appropriate sample preparation and data processing.*

Overall, the most significant limitations of this study should be addressed in some way. Detailed comments and recommendations on a range of aspects are provided below and this feedback will hopefully provide useful input that will guide the authors in further improving the manuscript.

*We hope we have addressed the reviewer's concerns.*

Specific comments:

L17: the use of 'blue reflectance intensity' is somewhat confusing and unnecessary. I would suggest using either the term 'blue reflectance' or 'blue intensity', or otherwise reword to 'blue intensity from reflected light'.

*So corrected – we now use "blue intensity" or "intensity" from this point through the paper.*

L25: In terms of response, DB results appear similarly mixed and temporally / seasonally variable when considering New Zealand and Tasmania sites on an individual basis. The stronger temperature response for Tasmania over New Zealand really only appears in the multivariate PCA results (Fig.5) – please amend the sentence to reflect this.

*Text adjusted to: "DB results were mixed, but performed better for the Tasmanian sites when combined through principle component regression methods than for New Zealand"*

L27 (& L424-425): This is somewhat misleading as the '52% to 78%' values refer to the (~50-yr) early / late split period calibration segments rather than the full period calibration results. The statement should instead include '55% to 64%', the temporal span of the calibration period should also be stated and the sentence reworded accordingly.

*These split period calibration values were detailed as they are associated with the validation (CE) values. Sentence has been tweaked to highlight that these values are for the split period: "Using the full multi-species/parameter network, excellent summer temperature calibration was identified for both Tasmania and New Zealand ranging from 52% to 78% explained variance for split periods (1901-1950 / 1951-1995), with equally robust independent validation (Coefficient of Efficiency = 0.41 to 0.77)."*

L32: WA should be defined in the abstract and other relevant sections of the text. I would also strongly suggest using the term quantitative wood anatomy along with the acronym (QWA) throughout the text as this would be more consistent with the literature.

*So changed: "Comparison of the Tasmanian BI reconstruction with a quantitative wood anatomical (QWA) reconstruction shows that these parameters record essentially the same strong high frequency summer temperature signal. Despite these excellent results, a substantial challenge exists with the capture of potential secular scale climate trends. Although DB, band-pass and other signal processing methods may help with this issue, substantially more experimentation is needed in conjunction with comparative analysis with ring density and QWA measurements."*

L45-46: 'unexpected' in what sense? Please specify / elaborate.

*This aspect of the sentence has been removed and we simply state: "However, the climatic influence on RW can be complex and there are many published studies where the relationship between RW and climate is shown to be temporally unstable and/or non-linear (Wilmking et al. 2020)."*

L50-51: While this is true, there are also limits to this and there are other considerations / limitations that can also affect density data (and by extension BI) – e.g., Divergence phenomenon.

*It is not clear what the reviewer is specifically referring to here. We simply state that MXD can portray a significant temperature response below tree-line well beyond where the same (weaker) response in RW data has faded completely. No change to the text made.*

L104-105: The last part of this sentence should be reworded since latewood density / LWB does not represent the density of the cell wall itself, but instead it reflects the proportion of cell wall to lumen area.

*As the reviewer is fully aware, there is still some debate as to what BI, QWA, density truly represent within the ring (see Kaczka and Wilson 2021). We have followed terminology from Björklund et al. (2019). However, to address the comment which is compatible with what we already state, we have tweaked the text to: "Based on ecophysiological theory (Buckley et al. 2018) we posit that EWB, derived from maximum intensity values of the whole-ring reflectance spectrum, essentially provides a surrogate for mean lumen size of the earlywood cells, while LWB, derived from minimum reflectance values, reflects the relative density (i.e. proportion of cell wall to lumen area) of the darker latewood cell walls."*

L105-107 / L435-436: I suggest reformulating these sentences as the statements may overstate the merits of BI, particularly when making generalizations based only on the specific context of this study. While the high frequency comparison of the BI and QWA chronologies is certainly encouraging, I would, however, urge more caution in comparing / equating the information that can be extracted using BI with wood anatomical data bearing in mind that QWA offers higher resolution (individual cell level), intra-annual information, an extensive range of parameters, etc. Since this study only examines the high-frequency component, I would expect QWA to have an additional advantage and an edge over BI in the development of reconstructions not restricted in this way, particularly considering that the utility of high-frequency reconstructions is limited. Furthermore, note that anatomical parameters can even exceed density / BI parameters in terms of climate sensitivity and the climatic response of other anatomical parameters such as maximum radial cell wall thickness (compared for example to mean cell wall thickness / cell diameter) may provide an even stronger signal (see e.g. Björklund et al., 2020).

*Please note that L105-107 is talking about QWA vs MXD and is not BI related. However, we now state that QWA climate response may be stronger than MXD: "The use of wood anatomical parameters for dendroclimatology has gained traction in recent years due to improvements in measurement methodologies allowing for the development of well replicated chronologies (Drew et al. 2013; von Arx et al. 2016; Prendin et al., 2017; Björklund et al. 2020). The strength of relationships between climate parameters and wood anatomical properties such as latewood cell wall thickness, tracheid radial diameter and microfibril angle is comparable to and can be stronger than maximum latewood density (Yasue et al., 2000; Wang et al., 2002; Panyushkina et al., 2003; Fonti et al., 2013; Allen et al. 2018)."*

*w.r.t. L435-436, we simply present comparative results between independent QWA/RW and BI/RW based multi-variate parameter reconstructions for Tasmania. At high frequencies they agree very well. In the context of the exploratory nature of the paper, we believe these results are extremely encouraging. We further provide substantial cautionary discussion about the potential limitations of BI data w.r.t. low frequency trends which cannot be addressed in this current paper but have been overcome in multiple studies in the Northern Hemisphere.*

L108-109: This statement indirectly suggests that some kind of evaluation of wood anatomical properties was performed, which is not the case. Presumably, the intention here is to use climate response info to pre-screen and highlight sites, which could be the focus of dendroanatomical / paleoclimatic research in the future? Please modify the statement to clarify this point.

*Yes – this was poorly worded – sentence now edited accordingly: "As well as undertaking a dendroclimatic assessment of multiple BI parameters from different Australasian conifers, our analysis will also identify which species would be a good focus for further BI and QWA measurement in the future."*

L111: what is meant by 'low performing'? Relatively poor / weak climate signal? Consider rewording.

*Sentence reworded to: "Improving terrestrial based estimates of past temperature in the land-limited Southern Hemisphere (Neukom et al. 2014) will only be achieved by enhancing the strength of the calibrated signal that until recently has been characterized solely by ring-width data which generally express a weak temperature signal."*

L118: It is not immediately clear that this means four species each from Tasmania and New Zealand (i.e. 8 species in total). Please modify for the sake of clarity.

*Reworded: "Four tree species from Tasmania and four from New Zealand were targeted for analysis (Figure 1, Table 1)……..". Including also the reference to Table 1, we believe that this should be clear now.*

L126-127: Can you specify which wood anatomical properties?

*Information now added: "By contrast, summer temperature calibration experiments performed on measurement series of several wood anatomical properties (e.g. tracheid radial diameter, cell wall thickness and microfibril angle), as well as RW and ring density, from these same species, have shown substantial improvements over RW alone (Allen et al. 2018), as well as the development of hydroclimate reconstructions (Allen et al. 2015a/b)."*

L133-134: In addition to ENSO, is there potential to capture / reconstruct past variability of any other atmospheric circulation phenomenon (e.g. Southern Annular Mode) using any part of this network? If so and if relevant, this could be mentioned in the discussion for example.

*There is no doubt that other climate dynamical phenomena have an important influence of the climate of this region – w.r.t. SAM – see for example:*

*Villalba, R., et al 2012. Unusual Southern Hemisphere tree growth patterns induced by changes in the Southern Annular Mode. Nature geoscience, 5(11), pp.793-798.*

*The Indian Ocean Dipole and Antarctic Oscillation (related to SAM) as well as many large volcanoes located in the Indo-Pacific Warm Pool region all influence climate variability across the study region – these will be the focus of our new NSF-NERC grant.*

*However – this paper's focus is on the fundamental local based climate signals (temperature and/or precipitation) expressed by these different tree-ring parameters. If they are robust local climate proxies, they will of course, be useful for larger scale dynamical studies. No changes made.*

L118-134: Most of this paragraph would perhaps be more suited for the introduction rather than the methods section.

*The section is "Data and Methods". At the moment, the Introduction is focused on introducing the different tree-ring parameters and why BI parameters might prove useful for enhancing RW based dendroclimatology in the region. We prefer for the information to stay in this section as it is a description of the "data".*

L146: Was the option to dismount and chemically treat the samples considered? What was the reason for not attempting this? Also, does the resin extraction of samples from only one site (AHA) influence the site / species comparisons in this study in any way?

*For the purpose of the current study, we never considered unmounting any samples and all samples were simply re-sanded to remove pencil marks, scanned and measured. From the outset, we planned to ONLY examine the high frequency signal of the generate BI data. In the early days of the BI method development, we employed this approach for Scots pine (Scotland):*

*Wilson, R., Loader, N.J., Rydval, M., Patton, H., Frith, A., Mills, C.M., Crone, A., Edwards, C., Larsson, L. and Gunnarson, B.E., 2012. Reconstructing Holocene climate from tree rings: The potential for a long chronology from the Scottish Highlands. The Holocene, 22(1), pp.3-11.*

*But also employed this initial approach for other tree species around the Northern Hemisphere. It is a quick efficient way for assessing the potential of a species for BI based dendroclimatology before "doing it properly".*

*In fact, for historical dating purposes, multiple experiments for Scots pine (Scotland) have shown that when the data are high pass filtered, resin extraction makes no difference w.r.t. the high frequency signal fidelity.*

*The fact that the AHA samples had been resin extracted likely had little difference to the results of the paper although we would freely admit that it would have been optimal to be consistent between all sites but that was not possible unless we ignored the already published (Blake et al. 2018) AHA data.*

L154: Please include a table in the supplementary information section specifying sanding grade, scanner type and scanning resolution for each dataset as well as which parameter settings were used to derive the EW / LW BI data (at least specify the percentage of light / dark pixels used for the EWB / LWB / DB calculation).

*We are not convinced that such a table would prove particular useful, but from a methods point of view, we admit that providing general information on the BI measurement settings would be useful. We have therefore added in this sentence: "Regardless of image resolution, the CooRecorder BI generation "window" was set to roughly equate to two thirds width of the sample while the window depth encompassed either the latewood or earlywood for each ring. The BI data were extracted following the method detailed in Buckley et al. (2018). For LWB, mean reflectance values were taken from the lowest 15% of the darkest pixels, while for EWB the mean of the brightest 85% of the pixels was used."*

L160: Specify the reason (i.e., different behavior of LWB in Australasian tree samples compared to Northern Hemisphere)

*We provide some discussion about this issue later in the paper. This is the methods section, and we simply state that we do not invert the LWB data as is the norm for NH conifers. In fact, the inversion of LWB data is only made to ensure that the LWB (inverted) data are compatible with standard detrending methods for MXD which often assume some sort of declining trend (raw LWB would show an increasing trend). This is not a relevant issue in the context of this paper as we use such a flexible spline for detrending.*

L172-173: The rationale for producing DB to try to mainly correct the heartwood / sapwood color transition issues is somewhat contradictory considering that at the same time flexible detrending and retention of only high-frequency variability is used here to bypass the color transition issue. In other words, this form of detrending may eliminate some or even most of the potential advantages of the DB parameter.

*The text is modified to state that we are assessing the high frequency potential of DB: "As no resin extraction was performed (except site AHA, Table 1) and all the species used for this study express a colour change from heartwood to sapwood, DB data will also be examined to explore its high frequency dendroclimatic potential." We reiterate that if the DB data does not provide a robust high frequency signal, then it really does not matter what its low frequency trends are.*

L196: It may be worth mentioning that the CRUTS 4.03 datasets start in 1901.

*Text so corrected*

L194-199: This paragraph should also include a statement mentioning that the instrumental series were filtered in the same way as the tree ring series.

*This information had been stated elsewhere, but the following text was added: "The climate data were similarly detrended as the tree-ring data to ensure consistency"*

L205-207: I am not entirely convinced that such an approach is adequate, particularly considering the relatively large distances involved. The good agreement could simply be due to a limited number of available stations used to produce the gridded dataset (particularly in earlier parts of the 20th century) rather than actual good agreement over these spatial scales. It may be helpful to examine the temporal availability of data and location of stations that contribute to the New Zealand gridded dataset (i.e. the three gridboxes used) over time. As spatial biases linked to the availability of instrumental data could be an issue, it may also be useful to perform the same comparison using only instrumental / remote sensing data from recent decades with better / more representative local spatial coverage to help substantiate this high degree of agreement. In any case, if appropriate, this type of limitation should also be mentioned in the methods or discussion sections.

*Please see earlier response – we believe we have addressed this issue in the context of the primary aims of the study*

L210 / L365-366 / Figure 5: Does it actually make sense to use the CE statistic in the context of this study considering that only the high-frequency component is explored? The issue is that the CE stats may appear more favorable with higher values compared to a calibration / verification assessment that would also contain lower frequency variance. The use of the CE statistic may therefore not be as stringent as it may seem considering that the means of the calibration and verification periods essentially do not differ in this case. In other words, even though CE is usually seen as more rigorous compared to the reduction of error (RE) statistic, in this particular case, RE would likely yield very similar values in this context and so it could just as well be shown instead of CE. This should be addressed and clarified in the text.

*Yes – the reviewer is correct – in fact, for these high passed reconstruction, validation r2, RE and CE are very similar. RE and CE are similar to the 3$^{rd}$ decimal place, that is why we only presented CE as the metric still represents a measure of validation coherence. It is not clear what the reviewer wants us to do, so no changes made as CE is provide a valid high frequency assessment of coherence.*

L223: Although EPS may arguably mostly reflect and perhaps be overly sensitive to high-frequency relationships, the replication requirements to meet the EPS = 0.85 threshold could potentially be even higher for unfiltered / less flexibly detrended series than the 20-yr spline detrended series. This may be worth mentioning.

*Or – the EPS could be lower. We do not know so we would prefer not to speculate and focus on the data and results we examine in the paper. Note that it is standard for EPS to be derived from sliding 30 or 50 year windows which actually minimised the overall assessment of the impact of low frequency trends on RBAR and EPS. No changes made.*

L257-258: Please specify / expand on why these results are 'extremely encouraging'.

*Sentence changed to: "Overall, the consistent and strong correlations of EWB with summer temperatures are extremely encouraging and show great promise for enhancing RW-based temperature reconstruction for both regions."*

L260: It may be helpful to include a reminder that this refers to non-inverted LWB data.

*Text changed to: "Both RCS (Celery Top) and high elevation Huon pine (MHP) express negative correlations that are in line with the positive MXD/temperature relationships noted in the Northern Hemisphere as the LWB data are not inverted"*

L261-262: Be careful with the formulation as the current wording may suggest that there is also a negative relationship between MXD / temperature. Consider rewording to avoid any confusion.

*See above – should be clearer now*

L260-270: The interchangeable use of site names / codes and species names is a bit confusing. Please be more considerate and consistent in their use throughout the text. For example in L261 either state the site code and species in brackets or vice versa, but not both. Also, L262-263 starts by referring to sites and the latter part talks about species.

*Although the details of the site codes and species are in Table 1, we now state the site code and species through the whole paper.*

L265: Could simplify to latewood anatomical parameters.

*So changed*

L268-270: Such behavior would typically indicate the influence of moisture limitation. Although the physiological behavior of Australasian species may indeed be very particular, I would still recommend also testing the response of chronologies using a relevant drought index (e.g., scPDSI) to check that growth is not reacting to moisture limitation (rather than just examining the response to precipitation).

*Correlation response function analysis was also undertaken against PDSI, but as the results were weak, we focussed on temperature and precipitation – the latter providing the potential for assessing whether the trees and the different parameters expressed moisture limitation in some way. No correlation with PDSI was stronger than correlations with either temperature or precipitation. There is no significant evidence of moisture limitation in these data.*

*We do defer to earlier statements, however, from both reviewers that we should provide more ecological information as to potential "expectations " w.r.t. temperature of moisture limitation. To keep the paper to a manageable size, we will not add in the PDSI results, but here is a short summary for the 1902-1995 full period analysis focussing on any significant correlations with an optimal season that is time-stable over both periods of analysis:*

*Tasmania*

*RCS: PDSI vs LWB and DB: r = 0.31 and -0.24 with DJ season. Stronger correlations with temperature: r = -0.34 and 0.39. No significant correlation with precipitation.*

*MCK: no sig correlation with PDSI or precipitation. Temperature signal is with EWB and LWB – see main paper.*

*CM: no sig correlation with PDSI or precipitation. Temperature signal is with EWB and LWB – see main paper.*

*MWWTRL: no sig correlation with PDSI or precipitation. Temperature signal is with RW, EWB and LWB – see main paper.*

*MRD: no sig correlation with PDSI or precipitation. Weak temperature signal is EWB – see main paper.*

*MHP: no sig correlation with PDSI. Weak inverse correlation between RW, EWB and DB with JFM precip but stronger correlations between RW, EWB and DB with summer temperatures - see main paper.*

*BUT: no sig correlation with PDSI and precipitation. Time instable correlation with summer temperatures for LWB (positive) and DB (negative) - see main paper.*

*New Zealand*

*PKL: weak correlation of LWB (-0.24) and DB (0.25) with DJ PDSI. RW correlation with DJFMA precip (r = -0.35). Much stronger correlations with temperature for RW (-0.43, DJFMA), LWB (0.57, NDJ) and DB (-0.44, DJFMA) - see main paper.*

*HUP: no sig correlation with PDSI, with some weak correlation with summer precip for RW (-0.36), EWB (0.28) and LWB (0.22). Temperature expresses strongest correlation with EWB but only r = 0.30 - see main paper.*

*FLC: no sig correlation with PDSI and precipitation. EWB and LWB correlation summer temperatures - see main paper.*

*AHA: weak inverse correlation of RW and DB with PDSI (r = -0.30 and -0.26 for DJF). No correlation with precipitation. Temperature returns significant positive correlation with RW and LWB and most strongly with EWB for DJ (r = 0.63) - see main paper.*

*DPP: No significant correlation with PDSI. Inverse correlation of EWB and LWB with precipitation (r = -0.45 (DJ) and -0.47 (DJF) respectively). Stronger positive correlations with temperature for EWB (0.65, DJF), LWB (0.44, DJFMA) and DB (0.53, DJ).*

*In summary – correlations with PDSI are either non-significant or weaker than other climate parameters. We believe a focus on the fundamental climate parameters of temperature and precipitation provide an adequate assessment of the climate response expressed by these data-sets. There is no gain to the understanding of the tree response of the study site by including correlations with PDSI.*

L285: Looking at the response of the EWB and LWB Pink pine (DPP) parameters, the seasonality is somewhat different, however both parameters are positively correlated with temperature and their seasonal response overlaps considerably (just broader / stronger for EWB), so the relatively high inter-parameter correlation is perhaps not that surprising.

*Sentence slightly modified to: "EWB and LWB data for the Pink pine DPP site express different early (Sep-April) and late (Feb-Apr) seasonal responses with temperatures (Figure 3), but still show a reasonably high inter-parameter correlation (0.60, Table A2) although this is partly expected as the response windows overlap."*

L291: 'Disappointing' in what sense? Weak / variable / inconsistent climate signal? Please specify.

*Have modified the sentence after this statement: "Overall, the DB results are mixed and disappointing. This parameter theoretically could minimise the colour bias of the darker to lighter colour heartwood/sapwood transition (Figure A1) but, for the data used herein, as the high frequency signal often portrays a mixed or weak signal with temperature, it suggests that the DB parameter might not be a valid approach to address the heartwood/sapwood transition bias."*

L291-294: I disagree with this statement since the very flexible detrending of the DB data in this study precludes the potential ability of this parameter to correct for the heartwood / sapwood transition (or any other multi-year color-related deviations). Hence, by only considering the high frequency, the full potential benefits of calculating DB could not really be assessed in this study and so this is a matter that will have to be investigated further. This also raises a broader point about the relevance of calculating DB while applying very flexible detrending more generally. Perhaps a more species-specific / individual site approach would be required to assess in more detail whether or to what degree DB can be helpful in the Australasian context. Please adjust the text with this in mind.

*See previous response. New text should better clarify this issue. Basically, if the high frequency signal of DB expresses a weak climate signal, then we cannot hope to use it to address the lower frequency biases of the HW/SW colour change.*

L297-298: I wonder whether the quality / spatial representativeness of precipitation data could play some role in this. In some cases, could the weak precipitation response possibly be partly related to poor local representation of instrumental precipitation in some locations?

*Yes, this is not impossible, especially for Stuart Island and western Tasmania. The maps show the longest precipitation records (> 100 yrs of data) for New Zealand and Tasmania. However, this would also be a relevant issue for gridded PDSI as well.*

[Figure]

*Previous work has found little moisture limitation response of these trees using RW data in these regions. As most of the target sites are from high elevation and the met stations are low elevation, and due to the rugged nature of western Tasmania, the reviewer highlights a very real problem which cannot be addressed in this paper. Due to the strong west-east precipitation gradient across Tasmania (very wet in west), it is very unlikely that the moisture limitation would be dominant in these trees. A similar situation exists for NZ although we admit that the lower latitude Kauri site may well be drier than their South Island counterparts.*

L307-308: It would be interesting to see if the response of DB would be even better compared to EWB / LWB if lower frequency trends were also considered.

*We struggle to see the relevance of this comment. If the DB data shows no high-resolution fidelity with climate, then it does not matter what low frequency signals we could extract from the data.*

L310: This probably suggests that rather than combining data from a wide range of species over an extensive spatial range, a more selective approach targeting specific species with similar parameter responses may be required in order to truly unlock the full potential of these datasets.

*We reiterate that the aim of the paper was to assess a range of BI parameters measured from a range of species – the assessment evolves from specific site and parameter analyses up to full network multi-variate combinations. The latter is no different than many large scale gridded dendroclimate products. What is important going forward is that a true eco-physical understanding of these relationships are explored and examined to better understand their utility of dendroclimate reconstruction. We fully agree that a focus just on EWB might well be the best approach for future work, but there is potentially also useful information in the other parameters. Our analysis simply highlights the potential of a multi-species/parameter approach. Our cautionary statement is perfectly valid in view of continuing research in this area. No changes made to text.*

L324-325: The degree to which this affects results could be tested. Would the response be just as strong or would it be closer to that of the other species if the two pencil pine sites were treated independently?

*There is no doubt that having species data from two sites will likely result in improved performance compared to only using single species. However, we refer the reviewer to the beta weight table in Table 2 which assess individual sites and parameters. The two EWB data-sets from the Pencil pine sites (MCK and CM) stand out which is not biased by the fact that there are multiple sites of the same species. No changes made to the presented analyses.*

L389: Is the elevation range of the Allen et al. dataset roughly similar to this study or are there major differences? Such factors could affect the calibration strength.

*Allen et al. (2018) utilised a larger number of sites covering roughly the same region as the study area in this paper as well as the same four species. The Allen et al. data-set is substantially better replicated than the data used herein. Text now modified to state this.*

L391-392: This implies that only the final reconstruction of Allen et al. was filtered with a 20-yr spline rather than detrending the individual series (as was done in this study). Could this affect the calibration strength in any way?

*As stated in the paper, we high passed filtered the Allen et al reconstruction with a 20-yr spline allowing those data to be directly comparable with the reconstruction developed in this study from 20-yr spline detrending TR series. We do not think this would affect the calibration at all.*

L397: The Berkeley temperature dataset could be introduced earlier in the manuscript (i.e., methods section).

*The Berkley data was used for the calibration of Allen et al, so feel that introducing these data at the point of comparing with Allen et al's reconstruction seems the better location in the paper.*

L402: Why not also detrend the Berkeley dataset with a 20-yr spline?

*The caption of Figure 6 states that the Berkley data were similarly high pass filtered ,although we note that the original caption had a typo and stated "linearly" instead of "similarly" detrended.*

L409-410: This is undoubtedly the most important factor. However, other issues that have not been examined may also be important and should be explored. The heartwood/sapwood issue does not necessarily represent the only color bias that may be relevant and so correcting only the heartwood / sapwood shift may not necessarily resolve all of the color-related issues in BI series.

*Sentence has been modified: "However, for their potential to be truly realised, the heartwood/sapwood colour change and other discolouration issues need to be overcome"*

L436-455: In time, perhaps more advanced imaging techniques can also be development to reduce or remove the influence of color on these datasets and so more effectively resolve such issues.

*We agree but do not want to speculate about "possible" advances. Our discussion focuses on current published approaches to date.*

L447-453: Such approaches could certainly represent an interesting solution to this problem. However, at the same time, the effectiveness of such techniques will need to be thoroughly evaluated and it is possible that such an approach may only be applicable to specific instances of color variation even in the context of heartwood/sapwood transitions (i.e. it may resolve some, but perhaps not all such issues effectively).

*We agree but we do not think there is anything wrong with our current text as we are discussing methods that have been employed for other similar problems.*

L455-457: Although again, factors such as elevation / latitude / distance from the treeline, the spatial representativeness of instrumental data, may play a significant role in the sense that mostly just one site per species (or two in a few cases) were evaluated and so it was not possible to evaluate whether the same species in a different location may respond differently. I believe this is an important point that should be mentioned.

*We have modified the text to: "At the very least, the results detailed herein, based on a limited number of sites per species, show that BI parameters can be used to identify those species that should be targeted for more costly and time-consuming analytical methods such as wood anatomical measurements.". Please note that the statement is specific to the utility of BI as a quick assessor or whether a site/species could be set aside for more time-consuming measuring using QWA methods.*

Figure 1: Are the coordinates for South / Central NZ correct (they should probably be switched)? The '°' symbols should also be corrected. However, I suggest adding the lat. / lon. info as a grid directly onto the maps instead.

*The lats/longs for the central and southern grids were the wrong way around – thanks for noting this.*

*So corrected: "Figure 1: Location map (basemap ESRI 2021) of the tree-ring sites used in this study (see Table 1). Also indicated (grey boxes) are the regional domains of the gridded CRU TS 4.03 temperature*

*and precipitation data (Harris et al., 2014) used for analyses. Tasmania: 145-147oE / 41-43oS; New Zealand: North: 173-175oE / 35-37oS; Central: 171-172oE / 42-43oS; South: 167-168oE / 46-47o.". We prefer to leave the coordinates in the caption and the superscript of "o" will presumably occur at the formatting step via the Journal.*

Figure 2: EPS info in panel B is difficult to see (please use larger font size). The figure shows that series with a higher CV generally also tend to have higher r-bar. However, this relationship is not as clear for the lower RBAR / CV range (i.e., the relationship within that lower range is considerably weaker and perhaps even absent. This could be pointed out more clearly in the text.

*Panel B EPS font increased in size as well as now red instead of grey.*
*The wider scatter of the EWB and LWB data points are already discussed in the text: "Although the range in RBAR values for the EWB and LWB data suggests some uncertainty in this observation (see also Table A1),………."*

[Figure]

Figure 2 / Figure 3 / Table A1: Table A1 clearly shows that the BI parameters are nowhere near EPS = 0.85, so how might this generally low replication affect results? Could this affect the representation of climate signal strength in some of these climate response assessments (particularly Figure 3 and A3) and since the replication seems to vary quite substantially (6-16 trees, 10-22 series), do more highly replicated sites have an advantage (i.e. stronger climatic relationships) and is it really a fair comparison? One possible way to check the potential impact of variable replication between sites would be to look at how the results change when all sites are limited to the same nr. of series.

*AS stated earlier, a high EPS value does not necessarily guarantee a strong climate signal. However, we believe the reviewer is missing the main point of the paper. We set out to evaluate whether BI parameters measured from a range of tree species from Tasmania and New Zealand could improve the dendroclimatic modelling based on RW alone. Despite the weak signal strength and purposely low replication, many of these data-sets express a strong high frequency climate signal and their multi-variate fusion through PC regression performs as well as a QWA derived reconstruction at these frequencies. It is for sure likely that with high replication, the results would improve. That is the focus of future work.*

Another related question is how and to what extent the results would differ if the EPS = 0.85 threshold was met by all chronologies? The point is that the lower replication introduces additional uncertainty in relation to the response results and so this should also be clearly stated as one of the limitations of this work.

*We believe we are very clear about the limitations of the current data-set. The conclusion clearly states, "Despite the weak common signal expressed by the BI parameters, the climate signal extant in these data*

*is very strong, especially EWB. When all parameters are combined using PC regression, depending on the period used, 52-78% of the summer temperature variance can be explained"*

More generally, could the relatively high replication requirements for some sites affect the attractiveness of BI in some cases (perhaps it would be easier to collect and process five samples and develop QWA series instead of 100+ BI series in some instances)? In any case, this is a matter that should be discussed.

*Based on the general weaker common signal expressed in many northern hemisphere BI chronologies (LWB or DB), the fact that more samples are needed to acquire an EPS of 0.85 has NOT been seen as a limitation. In fact, the speed/ease/cheapness of BI measurement more than makes up for the weaker signal strength as more samples can be measured until an EPS > 0.85 is attained.*

*Table A1 clearly shows that the number of series needed for some sites is not that large – for example, for Pencil pine EWB data (with associated strong climate signal) only 19 trees would be needed to attain an EPS of 0.85.*

*The reviewer also seems to forget that even with the weak signal strengths in these limited data, the calibrated r2 values are very good. All the BI parameters explain substantially more of the summer temperature variance than RW (except for DB in New Zealand – Figure 4A) despite the RW data expressing much stronger RBAR values (Figure 2). As the aim of the paper is to evaluate a range of species, the results provide adequate guidance for future work and potential target species.*

Figure 3: Consider adding Tasmania / New Zealand on the left side next to the site codes to more clearly distinguish the two groups of sites. The species names could also be included to help interpretation. Just as a suggestion out of curiosity, it would be interesting to see the persistence properties of the parameter chronologies (e.g. first order autocorrelation in the SI).

*We have inserted the following text into the caption of figure 3: "Upper block is for the Tasmanian sites, while low block is New Zealand. See Table 1 for site code names and species."*

*We did calculate $1^{st}$ order autocorrelation but the data did not make it into the paper. The values are of course sensitive to whether they are calculated from the raw chronologies or the 20-yr spline detrended chronologies. The table below details the $1^{st}$ order AC values. For the detrended data, the RW data portrays highest $1^{st}$ order AC, followed by EWB, LWB and then DB. However, we don't believe this information is relevant for the main aim of the paper however.*

| SITE code | raw | 2-yr spl | SITE code | raw | 2-yr spl |
|---|---|---|---|---|---|
| TASMANIA | | | NEW ZEALAND | | |
| RCSrw | 0.19 | -0.32 | PKLrw | 0.69 | 0.11 |
| RCSewb | 0.92 | 0.03 | PKLewb | 0.90 | 0.17 |
| RCSlwb | 0.58 | -0.22 | PKLlwb | 0.95 | 0.12 |
| RCSdb | 0.03 | -0.32 | PKLdb | 0.63 | 0.02 |
| MCKrw | 0.53 | 0.04 | HUPrw | 0.74 | 0.17 |
| MCKewb | 0.97 | 0.02 | HUPewb | 0.93 | 0.31 |
| MCKlwb | 0.93 | 0.10 | HUPlwb | 0.96 | 0.06 |
| MCKdb | 0.44 | 0.07 | HUPdb | 0.43 | -0.14 |
| CMrw | 0.52 | 0.11 | FLCrw | 0.72 | 0.41 |
| CMewb | 0.98 | 0.04 | FLCewb | 0.93 | 0.05 |
| CMlwb | 0.94 | 0.07 | FLClwb | 0.94 | 0.10 |
| CMdb | 0.62 | 0.08 | FLCdb | 0.65 | 0.24 |
| MWWTRLrw | 0.78 | 0.27 | AHArw | 0.56 | 0.11 |
| MWWTRLewb | 0.99 | 0.28 | AHAewb | 0.96 | 0.08 |
| MWWTRLlwb | 0.95 | 0.24 | AHAlwb | 0.97 | 0.17 |
| MWWTRLdb | 0.78 | 0.15 | AHAdb | 0.84 | 0.08 |
| MRDrw | 0.80 | 0.16 | DPPrw | 0.46 | 0.33 |
| MRDewb | 0.97 | 0.12 | DPPewb | 0.48 | 0.02 |
| MRDlwb | 0.90 | 0.18 | DPPlwb | 0.74 | 0.01 |
| MRDdb | 0.76 | 0.07 | DPPdb | 0.41 | 0.02 |
| MHPrw | 0.58 | 0.03 | | | |
| MHPewb | 0.80 | 0.07 | | | |
| MHPlwb | 0.78 | 0.07 | | | |
| MHPdb | 0.33 | -0.12 | | | |
| BUTrw | 0.87 | 0.18 | | | |
| BUTewb | 0.93 | 0.04 | | | |
| BUTlwb | 0.89 | -0.01 | | | |
| BUTdb | 0.30 | -0.10 | | | |

While differences in the response of Tasmanian sites are rather subtle and generally do not appear to be systematic, in contrast, the New Zealand sites may possibly express a gradient in the strength, seasonality, type of response (RW / EWB / LWB) from north to south (i.e., stronger / broader / later / seasonal response towards the south). This may be related to the large latitudinal spread of those sites and should be taken into account in the analysis and interpretation of the results. Furthermore, differences in site elevation and distance from the treeline between the New Zealand and Tasmania sites as a whole may also need to be accounted for, particularly considering that sites in Tasmania are generally located at high elevations compared to the much lower-elevation New Zealand sites. This could make it difficult to directly compare the two areas in terms of climate response as major differences in elevation may contribute to some of the observed differences in climatic response. These factors are not currently accounted for or discussed.

***We refer the reviewer to our earlier comment and text modification in the "Data and Methods" section that should now address this issue.***

Such factors along with the broad range of species examined could also partly be responsible for or contribute to the mixed DBI response (perhaps predominantly a reflection of the broad range of LWB response). This may indicate that the behavior of BI parameters should be examined in more detail within a more localized / species-specific framework. Overall, the role of elevation / latitude should be considered in this study.

***We fully agree but that is beyond the remit of the current paper.***

Figure 4: It is interesting that DB outperforms EWB in Tasmania, but performs very poorly when looking at the New Zealand sites. Does this perhaps suggest that it may be worthwhile to focus on different parameters when working with sites / data from New Zealand vs Tasmania (also keeping in mind that it might be more beneficial to use DB than EWB when lower frequency trends are also included)? It would also be helpful to show the PC loadings for the significant PCs based on the parameter / site variables used in Fig. 4 (A) and (B) as a supplementary table.

*A future strategy for sampling and further BI measurement will initially focus on the stronger sites we identified in this preliminary study – particular the Tasmanian Pencil Pine, King Billy pine and high elevation Huon pine which all represent "traditional" stands in temperature limited locations, as well as Silver and Pink pine from New Zealand. As discussed in the paper, signal processing methods to minimise colour bias will be developed. We have fully detailed that acquiring robust low frequency information from such data will be a challenge, but exploring the range of strategies employed in northern hemisphere studies including resin extraction and high replication will help substantially in better assessing DB or other signal processing methods to capture unbiased low frequency trends.*

*w.r.t. PC Loadings – the beta weights (Table 2) already provide the information we think the reviewer desires. These values denote the "weight" and relevance of the different variables in the full multivariate "space".*

Figure 5: It may be somewhat easier to navigate the statistical info presented in the figure if for example all ar2 stats were displayed at the bottom and CE at the top of the panels. The last approach (variant 3) appears to produce the most consistent pre-instrumental agreement compared to the other two variants. Is this an indication that it would perhaps be more appropriate to move forward with variant 3 rather than variant 2?

*Figure modified: ar2 values at the top, CE at the bottom.*

*The reviewer's latter comment is tricky to address as if we had used variant 3, another reviewer could easily have stated that we had statistically cherry picked the candidate input records for the PC regression. Variant 2 was used as the screening method (full period) as it is the most commonly used screening method in the wider paleoclimate community (now stated in paper).*

*There is in actual fact very little difference between the three full period calibrated variants and the main results of the paper would not change if we used the other variants*

[Figure]

Figure 6: Please add a 'correlation' or 'correlation coefficient' label next to the scale in the bottom right of the figure. Why not compare either using Berkeley or CRU rather than both datasets in this figure?

*Correlation coefficient label added.*
*CRU TS data were used for the time-series plots as these were the data used for the calibration trials in figure 5. The Berkley data were used for the spatial correlation as the comparison could be pushed back to 1841 – as stated in the figure caption.*

Figure 3 / Fig. A3d / Table A2: The climate response of DB appears to be related to its correlation with EWB / LWB. Higher correlations with EWB while low with LWB tend to yield a stronger positive relationship of DBI with temperature and vice versa. Considering the very low variance of EWB data (Table A2) and generally high negative correlation of LWB and DBI (and since only the high-frequency is retained), would it be reasonable to say that the DB chronologies are predominantly just an inverted version of the LWB chronologies? In relation to this point, I also wonder whether there is actually any real benefit to including the DB data in the PC regression / reconstruction exercise. In fact, presumably the full benefits of using DB are not apparent and cannot be assessed in this study since only the high-frequency is considered.

*Again – we reiterate that we have tested individual sites/parameters (Table 2, Figure 3) and multi-variate fusion of all parameters and species (Figure 4) and the full dataset (Figure 5). The most consistent response of the parameters is EWB with summer temperatures. The DB parameter could potentially correct for the heartwood/sapwood colour bias, but we have explored its high frequency climate signal and it has been found wanting. As stated in the paper, this is disappointing and although some useful relevant information could be extracted using DB from high elevation Huon pine (Tasmania) and silver/pink pine (New Zealand), the signal is still not as strong as EWB.*

*As for LWB and DB expressing similar information, but are simply anti-correlated with each other, the correlation response function results (Figure 3) and beta weights in Table 2 clearly show that their response is not that similar.*

Interestingly, the relationship between DB and RW seems to be rather strong for most sites (and stronger than RW vs LWB and EWB). Could there possibly be a RW size-related influence that could be affecting the calculation of DB (sensu Björklund et al., 2019)?

*Björklund et al., 2019 hypothesised that RW and LWB (not DB) would become more positively correlated with narrower rings and poor image resolution. We cannot test this hypothesis using the current experimental design and would rather not speculate. Again, we would reiterate that although RW and DB are often positive correlated (mean of 0.54 and 0.48 for Tas and NZ respectively), the correlation response function analysis in Figure 3 shows that the parameter response to temperature is rather different.*

Figure A1: It would make it easier to visually interpret the figure by including axis labels. Also, consider adding a vertical bar or series of vertical lines to show the position of the heartwood/sapwood transition (e.g., as in Fig. A2).

*Y axes are now labelled. Vertical lines have not been added to denote the HW/SW transition as the figure is already rather busy and for those species with a gradual colour change, adding in such a highlight is not practical.*

[Figure]

Minor comments:

L34: 'parameters' instead of 'variables'?
**Throughout the paper we have mostly used the term "parameter" rather than "variable". This appears consistent with the wider literature on BI.**

L35: 'properties' instead of 'features'?
**So changed**

L55: 'climate-limited'?
**Sentence modified to: "The reconstructive value of tree ring stable isotopes (carbon and oxygen) appears to be less constrained for sites where climate does not limit growth and substantial potential exists from mid-latitude regions where traditional dendroclimatological approaches are less reliable"**

L56: consider: '… where the application of traditional dendrochronological parameters is less reliable.'
**We have retained the original sentence as we want to focus on methods as well as variables.**

L69: consider adding 'considerable experience' to the list.
*Sentence modified to: "Despite the strong climate signal often noted in such non-RW tree-ring parameters, their procurement is expensive, often requires specialised equipment and experience, and is time consuming"*

L73: consider: 'MXD and BI essentially measure wood properties that are integrally linked'
*No change. We have retained the original sentence: "MXD and BI essentially measure similar wood properties."*

L83: This may simply be related to the way the font is displayed, however, please change 30oN to 30°N.
*We think this was a format imposed for the initial submission, but have changed through paper.*

L86: … at temperature-limited locations / sites
*Sentence modified and this is added: "from temperature limited sites"*

L92: delete 'the' in '… on the New Zealand's …'
*So modified*

L101-102: derived from maximum intensity values of the whole-ring reflectance spectrum
*Not clear what the reviewer is wanting here. No change made*

L108: change 'confers' to 'conifers'
*corrected*

L125: please change 'and has not been' to either 'and have not been' or 'and this species has not been'
*We used "have"*

L131 / L392 / L401: Please add '.' to 'Blake et al 2020' and 'Allen et al's'
*corrected*

L166-167: consider '… proposed a (statistical) procedure that could potentially correct …'
*Changed to "statistical"*

L185: 'more gradual' may be more appropriate than 'slower'
*Changed to "gradual"*

L226: Perhaps consider changing to 'The common signal is particularly weak for the following species: …'
*We retain the current wording as the sentence is also specific to BI variables as well, and the reviewer's suggestion would therefore not be optimal*

L245: Specify 'full 1902-1995 period'
*So modified*

L283: add 'respectively' at the end of the sentence
*No changes made – adding this word in would not improve the sentence form*

L297: Consider changing 'these chronologies' to something like 'these RW and BI chronologies' or 'the examined chronologies'

*Sentence modified to: "The previous section detailed that temperature is the predominant climate signal expressed across the Tasmanian and New Zealand RW and BI data studied herein"*

L303 / (L316): Consider changing to '… PCA identified three (RW), two (EWB), two (LWB) and two (DB) principal components …'
*Sentence modified to: "For Tasmania, the PCA identifies three (RW), two (EWB), two (LWB) and two (DB) principal components respectively"*

L329: 'Region-wide'?
*So changed*

L350: 'The Tasmanian modelling was performed against …'?
*So changed*

L366: 'ar2' should be defined somewhere
*Now defined although it is a standard regression metric*

L380: 'high-pass'
*So changed*

L387: 'regression-based'
*So changed*

L394 / 407: 'BI-based'
*So changed and throughout paper*

L425: 'higher' rather than 'greater'
*We prefer greater*

L432: It may be helpful to reword slightly simply to make it clearer that 'this inverse relationship' refers to the opposite response of SH sites with temperature compared to the relationship of NH sites with temperature (i.e. not an inverse relationship of SH site LWB with temperature).
*Slight modification to: "However, the relationship of LWB for most species with summer temperatures is opposite to that observed in the Northern Hemisphere and further study is needed to assess the physiological processes leading to this inverse relationship in these particular Southern Hemisphere conifers"*

L469-470: consider changing 'grey bars' to 'grey shading' and '1st / 2nd' to 'first / second'|
*So changed*

L475: change 'Heartwood/Sapwood' to lowercase (heartwood/sapwood)
*So changed*

L514: 'to undertake preliminary analyses'
*So changed*

Table 1: (caption) please change '5' to 'five'. (table heading last column) 'Period – 4 series' to something like 'Period > 3 series' or 'Period min. 4 series'
*So changed*

Figure 4 (caption) / L312: '… for all / individual parameters (A) and species (B).' Maybe also specify somewhere that the species calibration tests in (B) involve the full range of parameters.

*Caption changed to: "PC regression calibration (1901-1995) experiments for variables (all species) (A) and species (all variables) (B). A range of temperature seasonal targets are used with the strongest seasonal calibrations highlighted with circles."*